
# Functional flows for complex effective actions

Friederike Ihssen[1] and Jan M. Pawlowski[1,2]

**1** Institut für Theoretische Physik, Universität Heidelberg,
Philosophenweg 16, 69120 Heidelberg, Germany
**2** ExtreMe Matter Institute EMMI, GSI, Planckstr. 1, 64291 Darmstadt, Germany

## Abstract

In the present work we set up a general functional renormalisation group framework for the computation of complex effective actions. For explicit computations we consider both flows of the Wilsonian effective action and the one-particle irreducible (1PI) effective action. The latter is based on an appropriate definition of a Legendre transform for complex actions, and we show its validity by comparison to exact results in zero dimensions, as well as a comparison to results for the Wilsonian effective action. In the present implementations of the general approaches, the flow of the Wilsonian effective action has a wider range of applicability and we obtain results for the effective potential of complex fields in $\phi^4$-theories from zero up to four dimensions. These results are also compared with results from the 1PI effective action within its range of applicability. The complex effective action also allows us to determine the location of the Lee-Yang zeros for general parameter values. We also discuss the extension of the present results to general theories including QCD.

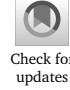

## Contents



# 1 Introduction

The phase structure of interesting relativistic quantum theories such as QCD, or non-relativistic ones in atomic and condensed matter physics such as graphene or spin-imbalanced fermionic gases, exhibits many interesting physics phenomena, ranging from critical end points to competing order regimes. In many cases these phenomena are related to the task of resolving complex structures in the theories at hand. Most prominently, this concerns partition functions with complex actions or Hamiltonians that typically lead to sign problems in a statistical approach. Moreover, constraints for the phase structure can be derived by considering complex external fields or parameters such as a complex magnetic field in spin systems. The latter extension gives rise to Lee-Yang zeros [1,2] in the complex (magnetisation) plane. These singularities restrict the radius of convergence of expansion schemes as well as providing at the same time much wanted information about the location of singularities on the real axis such as critical end points. For related works on the lattice, see [3–7], with the functional renormalisation group this has been studied in [8–11]. A further exciting possibility is the expansion of quantum field theories in trans series based on the expansion about complex saddle points.

These investigations require the computation of the partition function or free energy of the theories at hand at complex couplings or sources. In the present work we discuss functional renormalisation group approaches to complex action problems. The explicit applications in this work concentrate on the case complex sources, related to the Lee-Yang zeros mentioned above. We emphasise that the present complex action approach also accommodates quantum field theories with generic complex couplings. While also highly interesting e.g. in the context of PT-symmetric theories, explicit applications in this area will be discussed elsewhere. Here we only briefly discuss the minor technical differences within the numerical implementation of such flows.

We will argue that complex action flows are best formulated in terms of Wegner's flow equation [12] for general flows of an effective Hamiltonian or Wilsonian effective action, or in terms of the general flow for the 1PI effective action derived in [13]. These flows encompass the standard Polchinski equation [14] for the Wilsonian effective action and the Wetterich equation [15] for the 1PI effective action as specific cases. The generality is pivotal for setting up complex action flows adapted to the theory at hand.

In theories with complex actions and an intricate phase structure as discussed above, the solution of functional flows requires the use of advanced numerical methods. Moreover, different representations of functional flows may have advantages over others for specific problems, as they yield different types of parabolic equations. More specifically, we obtain parabolic partial differential equations reminiscent of convective heat equations for Wegner's flow [12], reaction-diffusion equations for the Polchinski flow [14], and hyperbolic equations in case of the Wetterich equation [15]. These types of equations find applications in a wide range of fields, such as electrodynamics, fluid mechanics and plasma physics. Combining features of both finite element and finite volume methods, Discontinuous Galerkin Methods (DGM) are particularly well-suited for solving this broad range of PDEs. Recent studies have successfully applied DGMs to the Wetterich equation in a real setting [16,17]. This work now expands these considerations to a more general RG-context in a complex setting.

Specifically we study the flow equation for scalar field theories in $d = 0$ to 4 dimensions with a complex source term. The $d = 0$ case can be solved exactly by other means and hence offers benchmark tests for our functional flows. We consider flows for the complex one particle irreducible (1PI) effective action, standard Polchinski flows for the Wilsonian effective action, as well as RG-adapted flows for the Wilsonian effective action derived from Wegner's flow equation. With the present formulation of these approaches, we find that the RG-adapted flows show best convergence in the complex plane. For the complex 1PI effective action we

set up and discuss a complex Legendre transform. We show that the results obtained from the complex 1PI flows pass the benchmark tests in $d = 0$. In higher dimensions it is compatible with the results for the Wilsonian effective action.

With the results for the effective potential for complex fields we determine the location of the Lee-Yang zeros as a function of the coupling parameters. We follow these locations towards their intersection with the real axis at the phase transition of the real theory.

We close this work with a discussion of extensions and further applications of the present setup for complex functional flows, in particular to QCD and the location of its critical end point.

## 2 Complex Functional Flows

The phase structure of theories with complex action parameters is quite intricate. In particular, the partition function may vanish at Lee-Yang zeros or exhibits cuts in the complex plane that start at the Lee-Yang zeros. As discussed in the introduction, in the present work we aim at setting up a functional approach that is flexible enough to generically deal with these structures. In the present Section we discuss general flows for the Wilsonian effective action (generating functional of amputated connected correlation functions) in Section 2.1, and the one-particle irreducible effective action in Section 2.2. These two generating functionals are related via a Legendre transformation, and the complex structure of the respective flows is different as are the types of the respective functional partial differential equations, see Section 4. In Section 3 the general setup is used to put forward an fRG flow that is adapted to the task of computing complex flows. The results in this work are computed within this flow, and other formulations are used as consistency checks and for comparison.

While the derivations in this Section and the following one, Section 3, apply to general theories, they are formulated in terms of a real scalar $\phi^4$ theory in $d$ dimensions for the sake of simplicity. The numerical results in Section 4 are also achieved for this theory in zero to four dimensions. In the smallest dimension, $d = 0$, the generating functional collapses to a simple one-dimensional integral and serves as a benchmark case. In turn, $d = 4$ is the critical dimension of the theory.

The classical action of the real scalar $\phi^4$ theory in $d$ dimensions is given by

$$S[\varphi] = \int_x \left\{ \frac{1}{2} \varphi(x) \left[ -\partial_\mu^2 + m^2 \right] \varphi(x) + \frac{\lambda}{4!} \varphi(x)^4 \right\}, \tag{1}$$

with the real scalar field $\varphi \in \mathbb{R}$, and the real mass and coupling $m, \lambda \in \mathbb{R}$. The $d$-dimensional space-time integral in (1) is abbreviated with

$$\int_x = \int d^d x \,. \tag{2}$$

The present formulation allows for the computation of complex effective actions deduced from complex masses and couplings, $m, \lambda$ as well as complex sources $J$ in the source term $\int_x J\varphi$. For the numerical application in the present work we consider consider complex sources $J$ and keep real masses and couplings. This allows for some numerical simplifications and covers the interesting case of Lee-Yang zeros. The more general situation, which occurs for example in applications to open quantum systems or PT-symmetric theories, e.g. [18,19], will be discussed elsewhere.

The starting point of our analysis is the path integral or partition function of the theory, $Z[J]$. All correlation functions in a Euclidean field theory can be obtained from this generating

functional. It is defined by its derivatives

$$\langle \varphi(x_1)\dots\varphi(x_n)\rangle_J = \frac{1}{\mathcal{Z}[J]}\frac{\delta^n \mathcal{Z}[J]}{\delta J(x_1)\dots\delta J(x_n)}, \tag{3}$$

which are the full normalised correlation functions including their disconnected parts. It also has a (formal) path integral representation,

$$Z[J] = \int d\varphi\, e^{-S[\varphi]+\int_x J(x)\varphi(x)}. \tag{4}$$

In the present work we shall consider complex currents $J$, which may be interpreted as a complex magnetic background field. In the presence of such a current and more generally also complex $m, \lambda$, the generating functional $Z[J]$ is also complex, as is the expectation value of the field and the higher correlation functions. However, it is a function of the complex variable $J$, and does not depend on its complex conjugate $\bar{J}$. Moreover, for real $m, \lambda$, the complex-valued generating functional $Z[J]$ and hence also the correlation functions are real functions of the complex variable $J$,

$$\overline{Z[J]} = Z[\bar{J}], \qquad m, \lambda \in \mathbb{R}. \tag{5}$$

While the generating functional $Z[J]$ in (4) generates the full correlation functions (3) including their disconnected parts, its logarithm

$$W[J] = \log Z[J] \tag{6}$$

generates the full connected correlation functions,

$$\langle \varphi(x_1)\dots\varphi(x_n)\rangle_J^{(c)} = \frac{\delta^n W[J]}{\delta J(x_1)\dots\delta J(x_n)}, \tag{7}$$

for a given background current $J$. Here, the superscript refers to the connectedness. Note, that for complex $Z[J]$, the logarithm in (6) introduces branch cuts.

Equation (7) includes the external propagators. They can be amputated by using a current $J = S^{(2)}\phi$, that is proportional to the classical dispersion

$$S^{(2)}[\varphi_0] = \frac{\delta^2 S}{\delta \varphi^2}[\varphi_0]. \tag{8}$$

This leads us to

$$S_{\text{eff}}[\phi] = -W[J = S^{(2)}\phi], \tag{9}$$

which defines the Wilsonian effective action. In (9) we have suppressed the background $\varphi_0$, typically chosen to be the vanishing background, $\varphi_0 = 0$. Derivatives w.r.t. $\phi$ lead to (7), where each field is multiplied by $S^{(2)}$, thus removing the classical external propagators. Moreover, the external current $J$ is now expressed in a background mean field $\phi$.

Finally, one-particle irreducible (1PI) correlation functions are generated by the Legendre transform of the Schwinger functional, the effective action $\Gamma[\phi]$,

$$\Gamma[\phi] = \sup_J \left[ \int_x J(x)\phi(x) - W[J] \right], \tag{10}$$

with

$$\langle \varphi(x_1)\dots\varphi(x_n)\rangle_\phi^{(1PI)} = \Gamma^{(n)}[\phi], \tag{11}$$

with a given mean field $\phi = \langle\varphi\rangle$. All generating functionals, (4), (6), (10), carry the full information about the theory under investigation with a decreasing degree of redundancy. Importantly, their functional flow equations constitute different general diffusion equations with different properties. This can be used to our advantage for the present task of solving them for complex effective actions.

## 2.1 Functional flows for the path integral measure

This endeavour is best started from the general flow equation for generating functionals, Wegner's flow equation [12]. There, general differential reparametrisation and RG-transformations of the theory at hand are considered. In terms of the Wilsonian effective action $S_{\text{eff}}[\phi]$ in (9), Wegner's flow reads

$$\partial_t P[\phi] + \frac{\delta}{\delta\phi(x)}\Big(\Psi[\phi]P[\phi]\Big) = 0, \qquad P[\phi] = e^{-S_{\text{eff}}[\phi]}. \tag{12}$$

The RG-time $t$ is the logarithm of the cutoff scale $k$,

$$t = \log k/\Lambda, \tag{13}$$

with some reference scale $\Lambda$. The cutoff scale can be both an infrared (IR) or ultraviolet (UV) cutoff scale or simply the (geodesic) parameter of a general reparametrisation [13]. In the present work we consider an infrared cutoff scale $k$ for explicit applications. This means that quantum fluctuations with $p^2 \lesssim k^2$ are suppressed below this cutoff scale and fluctuations with $p^2 \gtrsim k^2$ are integrated out (or in).

The exponential $P[\phi]$ in (12) is nothing but the path integral measure. General reparametrisations induced by (12) leave the path integral unchanged which is easily seen by integrating (12) over all fields: the integral of the right hand side vanishes as the integrand is a total derivative. The RG-kernel $\Psi[\phi]$ is typically chosen as

$$\Psi[\phi] = \frac{1}{2}\mathcal{C}[\phi]\frac{\delta S_{\text{eff}}[\phi]}{\delta\phi} + \gamma_\phi\phi, \tag{14}$$

with the boundary condition that $\Psi$ vanishes at $k = 0$ if $k$ is an infrared cutoff scale.

The second term on the right hand side of (14) with a field-independent $\gamma_\phi$ entails a rescaling of the field $\phi$, and has been introduced for convenience. Moreover, a field-dependent $\gamma_\phi$ can be considered as a driving term. Both, field-independent and field-dependent anomalous dimensions are commonly not considered in (14).

In turn, for $k \to \infty$ the kernel (14) should suppress all fluctuations, leading to a simple initial condition. These two requirements are more easily seen in terms of the flow for $S_{\text{eff}}$, for which Wegner's flow reads

$$\left(\partial_t + \int \phi\,\gamma_{\text{eff}}\frac{\delta}{\delta\phi}\right)S_{\text{eff}}[\phi] = \frac{1}{2}\operatorname{Tr}\mathcal{C}[\phi]\Big[S_{\text{eff}}^{(2)}[\phi] - (S_{\text{eff}}^{(1)}[\phi])^2\Big], \tag{15a}$$

where the field-independent term $\operatorname{Tr}\gamma_\phi$ was dropped and

$$\gamma_{\text{eff}}[\phi] = \gamma_\phi - \frac{1}{2}\frac{\delta\mathcal{C}[\phi]}{\delta\phi}. \tag{15b}$$

In (15a) we have also used the common notation

$$S_{\text{eff}}^{(n)}[\phi] = \frac{\delta^n S_{\text{eff}}[\phi]}{\delta\phi^n}. \tag{16}$$

The second line in (15a) is the trace of the RG kernel $\mathcal{C}[\phi]$ contracted with the connected two-point function of the theory. The first line contains the scale derivative of the Wilson effective action and a generalised anomalous dimension term. We note in passing, that we

may also recast the $(S_{\text{eff}}^{(1)}[\phi])^2$ term as part of the generalised anomalous dimension by shifting $\phi\,\gamma_{\text{eff}} \to \phi\,\gamma_{\text{eff}} + 1/2 S_{\text{eff}}^{(1)}[\phi]$ in the first line.

Standard Wilsonian RG-transformations are obtained for a field-independent kernel $\mathcal{C}$ with $\delta\mathcal{C}/\delta\phi = 0$, while field-dependent kernels introduce a reparametrisation of the theory. For more details and applications, in particular to gauge theories, see [20, 21], for a respective review see [22].

### 2.1.1 Standard flow for the Wilsonian effective action

Now we briefly describe how the standard Polchinski-type flow [14] for the Wilsonian effective action is derived from (12). Its derivation from the path integral is described in Appendix A. In short, we add an infrared cutoff function to the classical action of the theory,

$$S[\varphi] \to S[\varphi] + \frac{1}{2}\int_x \varphi R_k \varphi \,, \tag{17}$$

where $R_k$ is an infrared regulator and the RG-time $t$ as defined in (13). The regulator is typically defined in momentum space with

$$\lim_{p^2/k^2 \to 0} R_k(p) = k^2 \,, \qquad \lim_{p^2/k^2 \to \infty} R_k(p) \to 0 \,, \tag{18}$$

which also implies that $R_{k\to 0} \to 0$. Note, that in [14] an ultraviolet regulator was considered, but the structure of the flow is identical. However, infrared regulators directly implement the Wilsonian idea of integrating out momentum shells. Then, the RG-kernel is given by

$$\mathcal{C}[\phi_0] = -G_k^{(0)}[\phi_0]\,\partial_t R_k\,G_k^{(0)}[\phi_0], \tag{19}$$

and

$$\gamma_\phi = \partial_t G_k^{(0)}[\phi_0]S_k^{(2)}[\phi_0], \tag{20}$$

where the propagator $G_k^{(0)}$ is the classical propagator of the theory, including the regulator correction,

$$G_k^{(0)}[\phi_0] = \frac{1}{S_k^{(2)}[\phi_0]} = \frac{1}{S^{(2)}[\phi_0] + R_k} \,. \tag{21}$$

In the path integral this kernel is derived with the current

$$J = \left(G_k^{(0)}\right)^{-1}[\phi_0]\,\phi \,, \tag{22}$$

in the Schwinger functional $W[J] = \log Z[J]$, see (A.78) in Appendix A. When inserting this kernel in the general flow (15), the second line is simply the trace of $-1/2\partial_t R_k\,G_k[\phi]$ with the full propagator

$$G_k[\phi](x,y) = \langle \varphi(x)\varphi(y)\rangle - \phi(x)\phi(y), \tag{23}$$

with the mean field $\phi = \langle\varphi\rangle$, for more details we refer to Appendix A. In most applications one separates the full two-point function from the Wilsonian effective action,

$$S_{\text{eff},k}[\phi] = S_{\text{int},k}[\phi,\phi_0] - \frac{1}{2}\int_x \phi\,S_k^{(2)}[\phi_0]\,\phi \,. \tag{24a}$$

This split eliminates the trivial running of $S^{(2)}[\phi_0]$ from the flow and makes numerical computations more convenient. Inserting the split (24a) and the kernel (19) into the Polchinski flow (15a) leads to the flow of the interaction part $S_{\text{int},k}[\phi]$

$$\partial_t S_{\text{int},k}[\phi] = \frac{1}{2}\text{Tr}\,\partial_t G_k^{(0)}\Big[S_{\text{int},k}^{(2)}[\phi] - \big(S_{\text{int},k}^{(1)}[\phi]\big)^2\Big]\,, \tag{24b}$$

where we have dropped the $\phi$-independent term $1/2\,G_k^{(0)}\,\partial_t S_k^{(2)}$ on the right hand side, see also (A.84). The standard Polchinski equation is given by (24) with an ultraviolet regulator. We emphasise again, that the use of either UV or IR regulators makes no structural difference, while it does conceptually and practically.

Finally, it can easily be shown that for infrared cutoff kernels such as the Polchinski kernel (19), the decay properties (18) of the infrared regulator ensure a finite UV effective action as the initial condition. The flow of the UV relevant vertices is then governed by RG-consistency, [13, 23, 24].

## 2.2 General functional flows for the Effective Action

The 1PI analogue of Wegner's flow equation (12) for the Wilsonian effective action or effective Hamiltonians was derived in [13]. There, the starting point was the partition function with a source $\int J_\phi\,\hat{\phi}[\varphi]$ with the fundamental field $\varphi$, as well as a cutoff term for the composite field

$$S[\varphi] \to S[\varphi] + \frac{1}{2}\int \hat{\phi}[\varphi] R_k \hat{\phi}[\varphi]\,, \tag{25}$$

for the composite field $\hat{\phi}$, and possibly also cutoff terms for the fundamental fields. Then the Legendre transform is taken with respect to all the currents, including $J_\phi$,

$$\Gamma_k[\phi] = \sup_{J_\phi}\left(\int J_\phi\,\phi - \log Z_k[J_\phi]\right) - \frac{1}{2}\int \phi R_k \phi\,, \tag{26}$$

where we suppressed the potential Legendre transform w.r.t. the original field $\varphi$ for the sake of simplicity. For example, this general setup includes two-particle irreducible (2PI) actions (for $\phi(x,y) = \varphi(x)\varphi(y)$), and nPI actions, or density functionals (for $\phi(x) = \varphi(x)\varphi(x)$), see [13]. The general flow equation for such a 1PI effective action $\Gamma_k[\phi]$ with $\phi = \langle\hat{\phi}\rangle$ reads

$$\left(\partial_t + \int_x \dot{\phi}\frac{\delta}{\delta\phi}\right)\Gamma_k[\phi] = \frac{1}{2}\text{Tr}\,G_k[\phi]\partial_t R_k + \text{Tr}\,G_k[\phi]\frac{\delta\dot{\phi}}{\delta\phi}R_k\,, \tag{27a}$$

where $G_k$ is now the full propagator of the composite field $\phi$,

$$G_k[\phi](x,y) = \langle\hat{\phi}(x)\hat{\phi}(y)\rangle - \phi(x)\phi(y)\,, \tag{27b}$$

that is related to the inverse of the two-point function

$$G_k[\phi] = \frac{1}{\Gamma^{(2)}[\phi] + R_k}\,. \tag{27c}$$

The differential change $\dot{\phi}[\phi]$, is related to the expectation value of the differential variable transformation of the integration field $\hat{\phi}$ with

$$\dot{\phi}[\phi] = \langle\partial_t\hat{\phi}_k\rangle[\phi]\,. \tag{27d}$$

Equation (27d) defines the change of the composite field basis with the RG flow. We emphasise that $\phi$ itself does not depend on the RG-scale $k$, as it is the field/variable of the effective action. The change of the implicit dependence of the effective field $\hat{\phi}$ on the fundamental field $\varphi$ with the scale is *defined* via a given function $\dot{\phi}[\phi]$, for more details see [13], or a discussion of the special role of field zero modes and respective modifications see [25].

We remark that a variant of (27) has been derived in [26], based on (26) without regulators for the composite fields. Then the flow equation is simply the rotation of the standard flow equation in terms of the propagators of the composite fields, and hence the Jacobian of the transformation is accompanying all propagators. It can be seen as a special case of (27).

Wegner's flow equation for the Wilson effective action (12) and the general 1PI flow (27) are connected via the relation

$$\Psi = \dot{\phi}\,, \tag{28}$$

with an additional RG kernel $\Psi$. This can readily be checked with a Legendre transform, see also [27].

### 2.2.1 Standard flow for the 1PI effective action

We can reduce the general flow (27) to the standard flow equation of the 1PI effective action by using $\dot{\phi} = 0$. This choice entails that $\phi = \langle\varphi\rangle$ is the mean value of the fundamental field. Inserting this choice in (27) leads us to the Wetterich equation [15],

$$\partial_t \Gamma_k[\phi] = \frac{1}{2}\,\mathrm{Tr}\,G_k[\phi]\,\partial_t R_k\,, \tag{29}$$

see also [28, 29].

In summary, Wegner's flow (12) for the Wilson effective action (15) and its 1PI analogue (27) constitute the general functional flow framework that accommodates an adaptive setup of functional flows for complex actions: the kernel $\mathcal{C}$ or the transformation field $\phi$ can be adapted to the complex structure of the theory at hand.

## 3 RG-adapted flows

We now employ general RG kernels for the construction of *RG-adapted* flows. To begin with, we remark that the kernel of functional flows is given by the full field-dependent propagator $G_k[\phi]$, and hence any expansion scheme always implies also an expansion about $G_k[\phi]$. This property has been exploited in [30], and within conceptual considerations and applications on optimisation in functional flows, see [31–33]. In particular this led to functional optimisation as set up in [13, 23] as well as the recent development of essential RG flows [27, 34, 35]. This idea has also been picked up for Machine Learning applications to functional renormalisation in [36].

The above suggests to use RG kernels and currents that are constructed from the full field-dependent propagator. Here we briefly discuss a natural choice: In analogy to (22) we are led to the implicit definition,

$$J[\phi] = G_k^{-1}[\phi]\phi\,, \tag{30a}$$

and generalisations thereof. A respective RG kernel (12) is defined by

$$\mathcal{C}_k[\phi] = -G_k[\phi]\,\partial_t R_k[\phi]\,G_k[\phi]\,, \tag{30b}$$

and generalisations thereof. The latter generalisations are deduced from further optimisation conditions, that take into account the complex structure of the theory. Such an *RG-adapted* choice is very similar in spirit to the *dynamical RG* setup in [30]. The fully developed conceptual framework there will be very useful for the computational implementation, which is deferred to future work.

In this context we remark, that the implicit definition in (30) with the *field*-dependent propagator introduces a non-linear relation between the current and the field and requires an iterative solution of the flow. While it is precisely the non-linearity which is at the root of the optimisation, its practical use asks for a more comprehensive analysis. Hence, a full discussion of general optimised RG flows is deferred to a future publication. There we will also examine choices of (27d) and (30), that trigger stabilising positive diffusion terms (see Section 4) in either the flow of the Wilson effective action or the 1PI effective action.

## 3.1  RG-adapted expansion

In the current work we resort to a ready-to-use variant of (30): the theory is expanded about the full propagator on a fixed background. This leaves us with a linear relation between the current and the field,

$$J[\phi,\phi_0] = G_k^{-1}[\phi_0]\phi\,, \tag{31a}$$

with a field-independent infrared regulator $R_k$ and the respective RG kernel

$$\mathcal{C}_k[\phi_0] = -G_k[\phi_0]\,\partial_t R_k\,G_k[\phi_0]\,. \tag{31b}$$

For the background $\phi_0$ in (31a) a convenient choice is a solution to the equations of motion (EoM). The above definition (31) can be understood as an RG-improvement of (22): at each RG-step the respective full propagator is used to define the field $\phi$. This is an RG-adapted definition of the current or rather an RG-adapted expansion of the field.

With (31) we are led to the RG-adapted Wilsonian effective action

$$S_{\mathrm{ad},k}[\phi,\phi_0] = -W_k\big[G_k^{-1}[\phi_0]\phi\big]\,, \tag{32}$$

with the fluctuation field $\phi$, being the difference to $\phi_0$, for more details see Appendix B. From now on we suppress the dependence on the expansion point $\phi_0$ in (32) and simply write $S_{\mathrm{ad},k}[\phi]$. This definition entails an expansion of correlation functions and their flow about the full two-point function, and can be understood as an improvement in the sense of functional optimisation in [13].

The flow equation for the Wilsonian effective action $S_{\mathrm{ad},k}$ reads

$$\left(\partial_t + \int_x \phi\,\gamma_{\mathrm{ad},k}\,\frac{\delta}{\delta\phi}\right)S_{\mathrm{ad},k}[\phi]\frac{1}{2}\,\mathrm{Tr}\,\mathcal{C}_k\left[S_{\mathrm{ad},k}^{(2)}[\phi] - \big(S_{\mathrm{ad},k}^{(1)}[\phi]\big)^2\right],$$

with $\mathcal{C}_k$ provided in (31a) and the anomalous dimension

$$\gamma_{\mathrm{ad},k}[\phi_0] = -\big(\partial_t S_{\mathrm{ad},k}^{(2)}[0]\big)G_k[\phi_0] = \partial_t \log G_k[\phi_0]\,. \tag{33}$$

The generalised anomalous dimensions $\gamma_{\mathrm{ad},k}[\phi_0]$ is an operator and carries the change of the full (inverse) propagator with the cutoff scale. In any case, the flow equation (33) is well-defined for all complex fields $\phi$.

We note in passing that the use of the fully adapted kernel and current (30) leads to only minor modifications of (33): The form stays the same with the RG kernel $\mathcal{C}_k[\phi]$ in (30b) and

the modified anomalous dimension is given by

$$\gamma_k[\phi] = (\partial_t \log G_k) \frac{1}{1 + G_k \frac{\delta G_k^{-1}}{\delta \phi} \phi} - \frac{1}{2} \frac{\delta \mathcal{C}_k[\phi]}{\delta \phi}. \tag{34}$$

This eliminates any reference to an expansion field $\phi_0$ and the respective Wilsonian effective action only depends on the full field $\phi$.

The RG-adapted setup with the flow (33) admits a natural split of $S_{\mathrm{ad},k}$ in the kinetic part and the dynamical interaction part with

$$S_{\mathrm{ad},k}[\phi] = S_{\mathrm{dyn},k}[\phi] - \frac{1}{2} \int_x \phi \, G_k^{-1}[\phi_0] \phi + \mathcal{S}_0[\phi_0], \tag{35}$$

with the constant part

$$\mathcal{S}_0[\phi_0] = \frac{1}{2} \int_\Lambda^k \frac{dk'}{k'} \mathrm{Tr}\, \partial_t R_k G_k[\phi_0] + S_{\mathrm{ad},k}^{(2)}[0]\text{-terms}. \tag{36}$$

With the choice (36) we have $S_{\mathrm{dyn},k}[0] = 0$. Moreover, the RG-adapted field expansion entails that $S_{\mathrm{ad},k}^{(2)}[\phi_0]$ is (minus) the full inverse propagator in this background, see also (B.2). Hence, it follows from (35) that,

$$S_{\mathrm{dyn},k}^{(2)}[0] = 0. \tag{37}$$

Consequently, if we choose $\phi_0$ as a solution to the equation of motion with

$$S_{\mathrm{dyn},k}^{(1)}[0] = 0, \qquad \text{for} \qquad \phi_0 = \phi_{\mathrm{EoM}}, \tag{38}$$

the dynamical interaction part $S_{\mathrm{dyn},k}[\phi]$ is of order $\phi^3$ and indeed only carries interactions in an expansion about $\phi = 0$, and is an expansion about the physical mean field $\langle \varphi \rangle = \phi_0$, the full field being $\phi_0 + \phi$.

By inserting the split (35) into (33) we are led to the final form of the RG-adapted flow, which is also used for most of the numerical results in the present work. The flow for the dynamical part of the RG-adapted Wilsonian effective action reads,

$$\left( \partial_t + \int_x \phi \, \gamma_{\mathrm{dyn},k} \frac{\delta}{\delta \phi} \right) S_{\mathrm{dyn},k}[\phi] + \frac{1}{2} \int_x \phi \, \partial_t \Gamma_k^{(2)}[\phi_0] \phi$$
$$= \frac{1}{2} \mathrm{Tr}\, \mathcal{C}_k \left[ S_{\mathrm{dyn},k}^{(2)}[\phi] - \left( S_{\mathrm{dyn},k}^{(1)}[\phi] \right)^2 \right], \tag{39a}$$

with

$$\gamma_{\mathrm{dyn},k}[\phi_0] = -\partial_t \Gamma_k^{(2)}[\phi_0] G_k[\phi_0], \tag{39b}$$

where $\partial_t \Gamma_k^{(2)}[\phi_0]$ is the flow of the two-point function at $\phi = \phi_0$. This flow can be disentangled from that of the correlation functions $S_{\mathrm{dyn}}^{(n>2)}$, see Appendix B.

# 4 Numerical approach

In Section 2 and Section 3 we have discussed general flows for generating functionals. They are reminiscent of different types of (functional) partial differential equations (PDE), i.e. of parabolic and hyperbolic types. In the following, we will use this analogy to those types of PDEs to better understand the behaviour of the equations. We shall consider the flow of the generating functional or path integral measure (12), the Wilsonian effective action, (24) and (39), and the 1PI effective action (29). Our main results are achieved with the RG-adapted flow (39) for the Wilsonian effective action. We show in Section 5.2 that within the current approximation and the lack of RG-adapted reparametrisations for the 1PI effective action, the RG-adapted flow for the Wilsonian effective action is the most stable one for computing complex effective actions that originate from a complex source term. Thus we evaluate the stability of the other flows with this as a benchmark.

While the following considerations concerning the type of PDE are independent of the approximation, it is instructive to consider a simple approximation as a showcase example, the 0th order of the derivative expansion or local potential approximation (LPA). For a detailed discussion of this approximation scheme see e.g. [37]. This is also the approximation we will use in the numerics in Section 4. In short, the LPA only considers the classical dispersion in the generating functional under investigation and includes a full effective potential.

We start with the RG-adapted Wilsonian effective action, which is predominantly used for the numerical results in the present work. It is given by

$$S_{\text{ad},k}[\phi] = \int_x \left[ -\frac{1}{2}\phi(x)\left(-\partial_\mu^2 + m_k^2 + R_k\right)\phi(x) + V_{\text{dyn},k} \right], \tag{40}$$

where the dynamical part $V_{\text{dyn},k}$ of the effective potential only contains interaction terms, that is $\phi^n$ with $n \geq 3$ in a Taylor expansion about $\varphi = 0$.

The standard Wilsonian effective action in LPA is a variant of (40), where the full mass term with $m_k^2$ is frozen at $k = \Lambda$, and hence the remnant effective potential $V_{\text{int},k}$ also contains $\varphi^2$ terms,

$$S_{\text{eff},k}[\phi] = \int_x \left[ -\frac{1}{2}\varphi(x)\left(-\partial_\mu^2 + m_\Lambda^2 + R_k\right)\varphi(x) + V_{\text{int},k} \right]. \tag{41}$$

For the Wegner flow of the path integral measure we consider $\exp\{-S_{\text{ad/eff},k}[\phi]\}$ with the approximations (40) and (41) for the exponent. However, the complex Wegner flow has the issue of highly oscillatory initial conditions at high imaginary fields, see Appendix K for more details.

The 1PI effective action in LPA reads

$$\Gamma_k[\phi] = \int_x \left[ -\frac{1}{2}\phi(x)\,\partial_\mu^2\phi(x) + V_{\text{eff},k} \right]. \tag{42}$$

We emphasise again, that $\Gamma_k$ is the Legendre transform of the Wilsonian effective action. Therefore, the two effective potentials are not identical but are related by a Legendre transform. Hence, a given approximation of the respective generating functionals does not necessarily constitute the same approximation for the flows:

The $d = 0$ dimensional theory lacks the momentum dependence, and the LPA is exact. Hence, all flows have to agree with the full integral (after the Legendre transform is taken into account), if the initial conditions describe the same theory.

In turn, for $d > 0$ the LPA drops the non-trivial momentum-dependence of all terms. Accordingly, in LPA the Wilsonian effective action and the 1PI effective action differ genuinely.

In all cases the flow equations for the effective potential considered, (12), (24), (39), (29), can be formulated as convection-diffusion equations for the $\phi$-derivative of the effective potential or its interaction part,

$$u(\phi) = \frac{\partial V_{\text{dyn/int/eff}}(\phi)}{\partial \phi}. \tag{43}$$

With (43) the generic form of the functional flows is given by

$$\partial_t u(z) - \partial_z \left[ F(t, z, u(z)) - a(t, z, u(z))\, \partial_z u(z) \right] = 0, \tag{44}$$

where $t \in \mathbb{R}$ is the (negative) RG-time and $z \in \mathbb{C}$ is the complex field, triggered by a complex current or magnetisation in the path integral (4). In contrast to most of the DG literature, see [38–41], our formulation picks up an additional minus sign, due to the negative RG-time integration. As discussed there, all the generating functionals and hence their flows are real functions of a complex (field) variable $z$. The RG-adapted flow in LPA is derived in Section 4, see (56) and (57). The Polchinski flow and Wetterich flow in LPA can be found in Appendix A and Appendix H respectively.

The convection functional $F$ and the diffusion coefficient $a$ in (44) are structurally different in the functional flows considered here. Hence, already for real generating functionals, each of these systems of partial differential equations offers different numerical as well as conceptual challenges. Flows for complex generating functionals and the ensuing numerical evaluation of real generating functionals for complex fields add yet another layer of complexity.

In Section 4.1 we discuss the different types of partial differential equations encountered for the flows of the different generating functionals. In Section 4.2 we provide some details on the numerical approach with Discontinuous Galerkin (DG) methods. Finally, in Section 4.3 we discuss some features of the formation of DG for complex effective actions which allow to simplify the computation significantly.

## 4.1 Parabolic- and hyperbolic-type functional flows

Here we discuss the different types of partial differential equations (PDEs) we encounter for the three classes of functional flows put forward in Section 2 and Section 3. The emergent structure of singularities present in the different types of PDEs not only depends on the given type but also on the initial conditions. In the case of functional flows the set of allowed initial conditions is determined by RG-consistency [13, 24] and the physics at hand. This is discussed further in Section 5.1.2.

The flow of the path integral measure (12) is discussed in Section 4.1.1, that of the Wilson effective action (24) and the RG-adapted flow (39) are discussed in Section 4.1.2, and the flow for the 1PI effective action (29) is discussed in Section 4.1.3.

### 4.1.1 Parabolic-type flow I: Flow of the path integral measure

The Wegner flow of the path integral measure (12) with the typical RG kernel (14) is reminiscent of a *linear-parabolic equation*. Parabolic differential equations are common in heat conduction or particle diffusion processes. Generally speaking, real linear heat equations are well behaved. An initial solution $u(x, t_0)$ is smoothened out as the RG-time progresses and solutions exist for all times $t > t_0$. In an RG context, the diffusion coefficient is usually dependent on the RG scale. Thus, the smoothing can freeze out at a certain RG scale and a final structure survives. The amount of smoothing is therefore decided by the physical scales in the system which are introduced via the initial conditions.

Functional flows for the path integral measure on the real axis are showcased in Appendix K as a consistency check. In a complex setting, we find that the main intricacy in this formulation

are the initial conditions. As the complex part increases, the exponential function shows the characteristic oscillations which need to be resolved at a high numerical cost.

### 4.1.2 Parabolic-type flow II: Flow of the Wilsonian effective action

General flows for the Wilsonian effective action, and specifically the standard Polchinski flow for the effective action (24) and the RG-adapted flow (39) derived in Section 3 structurally resemble non-linear parabolic equations of the *reaction-diffusion* type. Subject to their specific form and the initial conditions, non-linear parabolic equations can generate so called *blow-ups* [40, 41]. These blow-ups may result in singularities, shocks or jumps at some finite time $t_0 < t_1 \leq \infty$. A prominent example for this flow is the Ricci-flow [42], which has been used to prove the Poincaré-conjecture.

We find that the initial conditions within the RG-setting belong to the class of initial conditions that potentially produce these blow-ups. Again, the RG scale dependence of coefficients can prevent the *blow up* by freezing out the system. Therefore, their occurrence or absence depends on the the details of the setup, in other words on the physics at hand. For a numerical investigation of blow-ups within the equations see Appendix F.

In the complex plane, this formulation is very similar to *two-component reaction–diffusion systems*. These types of systems are most prominently used to describe biological pattern formation, for a review see [43]. An interplay of differing diffusive contributions, as well additional source terms, can destabilise a homogeneous system, resulting in the formation of a periodic, static pattern [44]. This effect is primarily found in *activator-inhibitor systems* [45], which do not contain any convective contributions. However, these convective contributions can be found in RG-flows and are given by $F$ in (44). We therefore do not expect any static pattern formation.

Blow-ups in the lower dimensional ($d \leq 2$) solutions are directly related to the Lee-Yang zeroes [1, 2]. They are expected to show up as divergences in the Wilsonian effective action, simply by their definition as zeroes or cuts of the exponential. We demonstrate in Appendix F, that our RG-adapted flow (39) allows us to narrow down the position of the blow-up and leads to a quantitative estimate. Our present numerical scheme is not fully adapted for resolving such a singularity, and the resolution of this intricacy is subject of ongoing work in a fully RG-adapted setup.

In higher dimensions, $d > 2$, the Lee-Yang singularity is directly linked to a physical phase transition. Here, the singularity is related to a cut in the complex plane, which we cannot resolve in the present setup, as in the present approximation we enforce holomorphicity within the flow, see Section 4.2. Still, we are able to infer the position of the Lee-Yang in Section 6.2, since it lies at the beginning of the cut. This information is not tainted by our enforcement of holomorphicity. Numerical inaccuracies linked to a potential smudging out of a cut are also subject to further investigations.

### 4.1.3 Hyperbolic-type flow: Flow of the 1PI effective action

General flows for the 1PI effective action (27), including the Wetterich flow (29), are qualitatively very different from the previously discussed parabolic equation lookalikes. This qualitative difference is induced by the Legendre transform that connects the 1PI flows to that of the Wilsonian effective action and the path integral measure.

The 1PI flows are mostly dominated by strong convective movements and display characteristics of hyperbolic equations. Specifically, the information flows with a wave-like behaviour towards the infrared as has been studied in [16, 17, 46]. This structure of the information flow induces stabilising properties as does the dependence on the inverse 1PI two-point function. These stabilising properties are one of the main reasons why to date 1PI flows are used in

most numerical applications. However, 1PI flows display non-linear and even negative diffusive contributions, which appear at high densities, or large complex fields in the present setting. For an in depth analysis we refer to Appendix H.

### 4.1.4 Wrapup

The above investigation of the properties of the PDEs for the different functional flows suggests that the RG-adapted flows derived in Section 3 are very well-suited for the numerical computation of complex flows. The advantage is a combination of the structure of the PDE and the simple implementation of RG-adapted variable changes. We rush to add that this evaluation is based on the current state of the investigation of functional flows for complex effective actions. A full comprehensive investigation is deferred to a future work. The general setup put forward in the present work suggests that it is rather a combination of well-chosen initial condition and RG-adaptation that is important. We expect that within such a combination all different functional flows can be used equally well.

In summary, we will predominantly show numerical results obtained with the flow (39a) derived in Section 3 with the PDE-type discussed in Section 4.1.2. These results are also used as benchmark for the numerical results obtained with the other functional flows.

## 4.2 Discontinuous Galerkin

Our numerics is done with Discontinuous Galerkin methods, for an introduction see e.g. [38]. They have been used for a wide range of hyperbolic, elliptic and parabolic partial differential equations, e.g. [47–49], also in the context of blow-ups, e.g. [50]. Previous fRG works [16,17, 46] with discontinuous Galerkin methods, focused on the Wetterich equation, and hence the focus was on solving wavelike, convection dominate flows, for fRG works with related finite volume methods see [51–54].

The direct Discontinuous Galerkin method from [17] is well-suited for solving systems with dominant first order terms and wave-propagation processes. The respective flows converge using explicit time stepping schemes. However, in non-linear parabolic equations diffusive contributions play a big role. Therefore, in the present work we use the Local Discontinuous Galerkin method (LDG) [39] and implicit time stepping. This method has been set up for the fRG, and the corresponding numerical framework using the DUNE-project [55–59] can be found in [46].

In the present work we extend this method to complex systems of flow equations. The explicit form of each of these equations has already been displayed in (44). There we have introduced the complex field variable $z = x + iy$, which can be interpreted as a 'spatial' variable for the given type of PDEs. The form (44) is essential for the convergence of the LDG method. Importantly, $u, F \in \mathbb{C}$ are real functions of a complex variable $z$. This leaves us with a two component system of PDEs for two one-dimensional variables $t, z$. It is discussed in detail in Section 4.3 how this property facilitates the numerical implementation. In short, the one-dimensional spatial coordinate is resolved within a numerical approximation, while the time dependence is integrated via an implicit time stepping scheme.

The requirement $a \geq 0$ ensures a strictly positive diffusion, which is a necessary requirement for the convergence of the LDG method [39]. Positive diffusion is a very restrictive requirement on the general form of a system of PDEs. In particular, this requirement is not always met in the complex 1PI flows. In Appendix H the instability of a naive implementation of the 1PI flow in a complex setting is demonstrated numerically. In turn, the RG-adapted flow (39) was constructed with this property in mind. Lastly, the numerical implementation is outlined in Section 4.3.

### 4.3 Complex structure

The present set-up and in particular the final flow equation (44) can be readily applied to complex action problems with general complex couplings. Then, (44) is a partial differential equation for a general complex function $u(z)$ of a complex variable $z$. For complex classical couplings we find

$$\overline{u(z)} \neq u(z). \tag{45}$$

Accordingly, (44) has to be solved for its imaginary and real part, while also implementing the holomorphicity constraint $\partial_{\bar{z}} u(z) = 0$.

From now on we restrict ourselves to the case of complex currents. Then, (44) is a partial differential equation in the RG-time t and the complex spatial variable $z$. In principle, one could simply use the split of $z$ into its real and imaginary part, $z = x + \mathrm{i}\, y$, and the respective split of the derivative, $\partial_z = \frac{1}{2}(\partial_x - \mathrm{i}\, \partial_y)$, for solving the equation on a two-dimensional grid. However, this is a two-dimensional representation of a one-dimensional system, additionally necessitating the implementation of the holomorphicity constraint.

Naturally, it is much more efficient for the numerical implementation to utilise the complex structure: the generating functional (4) and all derived quantities are defined for real field variables $\phi$. They become complex with $\phi \in \mathbb{R} \rightarrow \phi \in \mathbb{C}$. Thus, we can exploit the fact, that we compute real functions of a single complex variable $z$. For the generating function this leads to (5). For a general real function $f(z)$ this simply reads

$$\overline{f(z)} = f(\bar{z}), \tag{46}$$

with the real and imaginary part satisfying

$$\mathrm{Re}\big[f(x,y)\big] = \frac{1}{2}\big(f(x,y) + f(x,-y)\big),$$

$$\mathrm{Im}\big[f(x,y)\big] = \frac{1}{2\mathrm{i}}\big(f(x,y) - f(x,-y)\big). \tag{47}$$

The Cauchy-Riemann differential equations allow to reformulate the $z$ derivative as a derivative of only the real variable $x$,

$$\partial_z \mathrm{Re}\big[f(z)\big] = \frac{1}{4}\Big(f^{(1,0)} + \overline{f^{(1,0)}} - \mathrm{i}\,\big(f^{(0,1)} - \overline{f^{(0,1)}}\big)\Big)$$

$$= \frac{1}{2}\Big(f^{(1,0)} + \overline{f^{(1,0)}}\Big),$$

$$\partial_z \mathrm{Im}\big[f(z)\big] = \frac{1}{4\mathrm{i}}\Big(f^{(1,0)} - \overline{f^{(1,0)}} - \mathrm{i}\,\big(f^{(0,1)} + \overline{f^{(0,1)}}\big)\Big)$$

$$= \frac{1}{2\mathrm{i}}\Big(f^{(1,0)} - \overline{f^{(1,0)}}\Big), \tag{48}$$

where we used $\partial_z = \frac{1}{2}(\partial_x - \mathrm{i}\partial_y)$. The validity of the Cauchy-Riemann equations is assumed on the computational grid, which does not contain the blow-ups. Note that this assumption artificially smoothens out non-analyticities (cuts) around critical regions, since it enforces holomorphicity.

In summary we can replace $\partial_z \rightarrow \partial_x$ without loss of generality if the solution is holomorphic. After this replacement (44) does not contain any dependency on $\partial_y$, which allows for the computation at constant $y$. Therefore, the complex plane can be resolved on slices of constant $y$, i.e. a one-dimensional numerical grid in $x$-direction. The $y$-slices are then put together

Table 1: Summary of the different schemes used throughout this work. We indicate the Effective Action (EA), the Effective Potential which is used, as well as the respective flow equation, and definition of the current.

| Scheme | Effective Action (EA) | Effective Potential | Flow eq. | Current |
|---|---|---|---|---|
| *RG-adapted flow* | RG-adapted Wilsonian EA (32) | Dynamical pot. $V_{\text{dyn}}$ (40), mass $m_k^2$ (52) | (56), (53) | (30b) |
| *Polchinski flow* | Wilsonian EA (9) | Interaction potential $V_{\text{int}}$ (41) | (G.2) | (22) |
| *1PI flow* | 1PI EA (10) | Effective potential (42) | (H.2) | (C.5) |

after the computation and we interpolate between them. Note that this procedure allows to resolve the entire RG-time and $x$ dependence for $\text{Im}[z] \neq \text{Im}[z_c]$, where $z_c$ is the position of the blow-up, whereas the computation of the $\text{Im}[z_c]$ slice freezes in at $t_1 < \infty$. This separation also illustrates nicely how the extension of the formalism to the complex plane does not affect the computation on the real axis.

### 4.4 Separating the equations

The real and complex part of (44) are now separated, and we use the replacement $\partial_z \to \partial_x$ as suggested by the previous Section. Here, we solely focus on parabolic-type flows, where the diffusive contribution is completely independent of $z$ and thus a real function $a(t) \in \mathbb{R}$. In the LDG formulation, this translates to system of two instationary and two stationary equations,

$$\partial_t u_x - \partial_x \left( \text{Re}\left[ F\left( t, z, u_x, u_y \right) \right] - a(t)\, q_x \right) = 0\,,$$
$$\partial_t u_y - \partial_x \left( \text{Im}\left[ F\left( t, z, u_x, u_y \right) \right] - a(t)\, q_y \right) = 0\,, \tag{49}$$

and

$$q_x = \partial_x u_x\,, \qquad q_y = \partial_x u_y\,, \tag{50}$$

where we used the notation $u = u_x + \mathrm{i}\, u_y$ and a negative RG-time $t$. Alternatively, one could have used the replacement $\partial_z \to -\mathrm{i}\,\partial_y$. The proof for this replacement works analogously to Section 4.3. This procedure maps $\partial_z^2 \to -\partial_y^2$, effectively changing the sign of the diffusion term.

In principle, a mapping of $\partial_z$ to a mixed expression of $\partial_x$ and $\partial_y$ is also possible. In this case the computation would be performed on certain trajectories in the complex plane instead of straight lines.

Applying the LDG formulation to the 1PI flow is much more challenging. The diffusion coefficient $a(t, z, u(z))$ is not only field-dependent and thus requires an additional numerical flux, but it is also complex and hence creates mixing terms. For a formulation of the 1PI LDG formulation in a real setting see [46]. The complex LDG 1PI setup is outlined in Appendix H.

## 5 Convergence in the complex plane

In this Section we present numerical results for the effective potential of the $\phi^4$ theory with a (classical) real scalar field and the action (1). The RG-adapted flow equation for the dynamical part $V_{\text{dyn}}$ of the effective potential is derived in Section 5.1, including a discussion of the initial conditions. The derivation of the standard Polchinski flow and the 1PI Wetterich flow are

deferred to Appendix G and Appendix H respectively. These Appendices also include further numerical details and results.

A first benchmark test is given by the computation of the effective potential in $d = 0$ in Section 5.2. In zero dimensions the effective action agrees with the effective potential and LPA gives the full result. Hence, this analysis provides us with a non-trivial numerical benchmark for the convergence of the different flows. As a next step, the RG-adapted scheme is investigated on the real axis and compared with flows for the effective Wilsonian action, as well as the 1PI results.

## 5.1 Flow of the dynamical potential

As already discussed before, we consider the 0th order derivative expansion or local potential approximation (LPA), in which the full effective action is approximated by a classical dispersion term and a full effective potential. This is a low momentum approximation and is well-tested, see e.g. the recent review [37] for a comprehensive overview. In LPA the Wilsonian and 1PI effective action take the form (40), (41) (Wilsonian), and (42) (1PI). For the convenience of the reader we have provided a tabular summary of the different schemes, and where to find them, in Table 1.

### 5.1.1 Flow equation

In this Section we apply the RG-adapted scheme from Section 3 to the real scalar field theory in $d$-dimensions. Solving (39a) for a single current $J = G_k[\phi_0]\phi$ requires knowledge of the full propagator at the expansion point $\phi_0$, which is given by

$$G_k[\phi_0] = \left(\Gamma_k^{(2)}[\phi_0](p) + R_k(p^2)\right)^{-1} = \left(m_k^2 + p^2 + R_k(p^2)\right)^{-1}, \qquad (51)$$

where the regulator $R_k$ is given by a flat cutoff, that is optimised for the 0th order in the derivative expansion, [13,31,32], see Appendix L.

In the RG-adapted scheme, the two-point contribution at the expansion point $\phi_0$ is separated from the field-dependent dynamical potential $V_{\mathrm{dyn}}$ (40), see also (37). Thus, in the RG-adapted scheme we have to solve two distinct equations. At every RG-time step, we first compute the two-point contribution at the expansion point, which is just the RG-time dependent mass $m_k^2$. Secondly, we solve the field-dependent flow of the dynamical potential $V_{\mathrm{dyn}}$. To derive the flow of RG-time dependent mass, we use its direct link to the 1PI two-point function,

$$m_k^2 = \Gamma_k^{(2)}[\phi_0](p)|_{p=0}, \qquad (52)$$

which in turn can be inferred from the flow of the RG-adapted effective action. This is evaluated in Appendix B by using the relations for the two-point functions, (B.2), and the flow of $\Gamma_k^{(2)}$ in terms of the RG-adapted effective action, (B.4). The latter is now used to obtain the flow of the RG-time dependent mass,

$$\partial_t m_k^2 = \frac{1}{2} \int_p \left\{ \frac{\left(V_{\mathrm{dyn}}^{(3)} V_{\mathrm{dyn}}^{(1)} - V_{\mathrm{dyn}}^{(4)}\right) \partial_t R_k(p^2)}{\left[m_k^2 + p^2 + R_k(p^2)\right]^2} \right\} = \frac{v(d) k^{d+2}}{\left(m_k^2 + k^2\right)^2} \left(V_{\mathrm{dyn}}^{(3)} V_{\mathrm{dyn}}^{(1)} - V_{\mathrm{dyn}}^{(4)}\right), \qquad (53)$$

where

$$v(d) = \frac{2\pi^{d/2}}{\Gamma(d/2)d}(2\pi)^{-d}. \qquad (54)$$

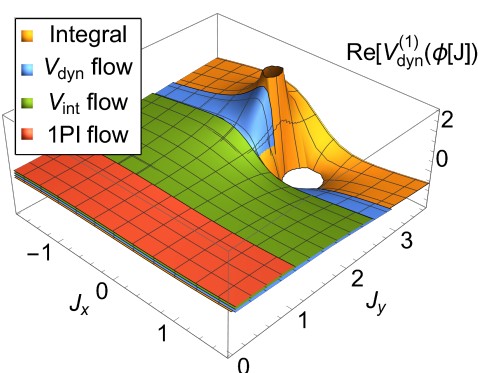
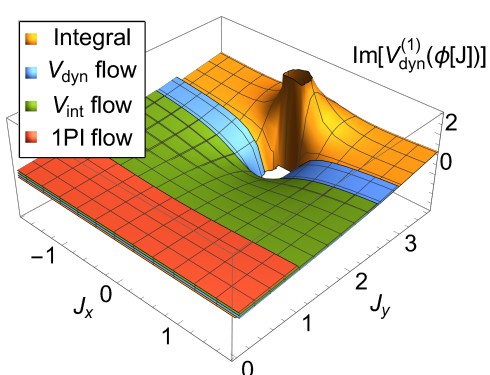

(a) Graphical comparison of the real part of $V_{\text{dyn}}^{(1)}$.

(b) Graphical comparison of the imaginary part of $V_{\text{dyn}}^{(1)}$.

Figure 1: Numerical convergence in the complex plane for the different expansion schemes. The results of all flows (*red*: 1PI, *green*: Polchinski ($V_{\text{int}}$), *blue*: RG-adapted ($V_{\text{dyn}}$)) agree with the results of the numerical integration, and for a better visualisation the different data are shown with a slight offset. We show results for the first derivative of the dynamical potential $V_{\text{dyn}}^{(1)}$, which is raw output of the numerical computation in (56). The convergence of the 1PI flow is analysed in Appendix H. Further definitions and fluxes of the different schemes are summarised in Table 1. All units are given in terms of the UV mass $m = 1$ with an initial cutoff $\Lambda = 5$.

In (53), the nth derivatives

$$V_{\text{dyn}}^{(n)} = V_{\text{dyn},k}^{(n)}[\phi_0] \tag{55}$$

are the corresponding n-point function of the RG-adapted effective action at the expansion point $\phi_0$. The full solution of $V_{\text{dyn},k}$ is computed within the Local Discontinuous Galerkin method. This method captures the field dependence on $\phi$ using non-overlapping cells. Within each cell $V_{\text{dyn},k}$ is then projected onto a higher order polynomial basis $N > 4$ (where $N$ is the polynomial order) around the expansion point. This basis ensures a precise computation of derivatives. Hence we can infer all vertices up to the four-point functions from the field-dependent potential with a very high numerical precision. The extraction of higher order derivatives from the full solution is expanded on in Appendix E.

The dynamical potential is a real function of a complex variable. We pick $\phi_0 = 0$ on the real axis, in order to ensure that this property is not spoiled by out choice of expansion point $\phi_0$. Thus all vertices $V_{\text{dyn}}^{(n)}$ are also real, implying $m_k^2 \in \mathbb{R}$. This reasoning can be extended even further by making use of the $Z_2$ symmetry of the effective dynamical potential $V_{\text{dyn},k}(\phi)$. We deduce that $\text{Re}[V_{\text{dyn},k}(\phi)]$ is an even function in the real variable $\phi_x$, whereas $\text{Im}[V_{\text{dyn},k}(\phi)]$ is odd in $\phi_x$. Furthermore, we find that $\text{Re}[V_{\text{dyn},k}^{(1)}(\phi)]$, $\text{Im}[G_k^{-1}[\phi]]$, $\text{Re}[V_{\text{dyn},k}^{(3)}(\phi)]$, $\text{Im}[V_{\text{dyn},k}^{(4)}(\phi)]$ are odd in $\phi_x$ as well. This implies purely real or imaginary values, for even and odd vertices respectively, if the expansion point is chosen at $\phi_x = 0$. Since odd vertices only appear as a product, we find that choosing any expansion point along the imaginary axis does not introduce any complex parts in the flow equation. By also choosing $\phi_0 = 0$, we deduce $V_{\text{dyn}}^{(3)} = V_{\text{dyn}}^{(1)} = 0$. Accordingly, the respective terms in the equations are dropped in the following.

Next, we derive the field-dependent flow of the dynamical potential $V_{\text{dyn}}$. For this purpose, (40) is inserted in the flow (39a). Using the expression for the propagator in (51) with the

flat cutoff, we arrive at

$$\partial_t V_{\mathrm{dyn}}(\phi) = v(d)k^{2+d}\left[\frac{\left[V_{\mathrm{dyn}}^{(1)}(\phi)\right]^2 - V_{\mathrm{dyn}}^{(2)}(\phi)}{\left(m_k^2 + k^2\right)^2}\,\frac{V_{\mathrm{dyn}}^{(4)}}{\left(m_k^2 + k^2\right)^2}\,\frac{\phi^2}{2} - \frac{V_{\mathrm{dyn}}^{(4)}}{\left(m_k^2 + k^2\right)^3}\,V_{\mathrm{dyn},k}^{(1)}(\phi)\phi\right]\,. \tag{56}$$

We now proceed to adjust this equation to the numerical framework presented in Section 4.2 and map $\partial_\phi \to \partial_{\phi_x}$. Equation (56) can be reformulated as a one-dimensional non-linear diffusion equation by taking an additional $\phi_x$-derivative. In a complex framework we make use of Section 4.3. With $u = \partial_{\phi_x}V_{\mathrm{dyn}}(\phi)$ as defined in (43), we obtain

$$\partial_t u = A(k, d, \phi_0)\,\partial_{\phi_x}\Big(F[u, \phi, \phi_0, k] - \partial_{\phi_x}u\Big). \tag{57a}$$

The real, positive diffusion coefficient is given by

$$A(k, d, \phi_0) = v(d)k^{2+d}\,\frac{1}{\left(m_k^2 + k^2\right)^2}\,, \tag{57b}$$

and the complex valued convective flux

$$F[u, \phi, \phi_0, k] = u^2 + u'''(\phi_0)\left[\frac{\phi^2}{2} - \frac{1}{\left(m_k^2 + k^2\right)}\,u\,\phi\right]. \tag{57c}$$

This highlights a convenient property of the RG-adapted expansion for the evaluation of complex effective potentials or even effective actions: for any expansion point along the imaginary axis the propagator remains real, i.e. positive diffusion is ensured.

### 5.1.2 Initial conditions

All dimensionful quantities are measured in powers of the mass at the initial UV scale $k = \Lambda$, in particular implying $m = 1$. The initial or bare classical coupling at $k = \Lambda$ is also set to unity, to wit

$$m^2 = 1\,, \qquad \lambda = 1\,. \tag{58}$$

The initial conditions are obtained via the Legendre transformation of the classical action, which is the initial condition for the 1PI flow. Details of this derivation are given in Appendix C. Furthermore, the choice of the initial cutoff scale $\Lambda$ in the UV requires special care: the algebraic structure of the Wilsonian effective action flow has no built-in suppression at high field values, as present for the 1PI flows. In fact, the flow of couplings and the mass increases at larger field values. Moreover, the structure of the flow is such that for $d > 2$ potential numerical inaccuracies in this fine-tuning problem are enhanced by powers of the cutoff scale.

In this work, an initial cutoff scale $\Lambda = 5$ is used, for which we find coinciding results from various methods for dimensions $d = 0, \dots, 4$, see Section 5.3.

## 5.2 Benchmark results in d=0

In this Section, the numerical convergence of the different functional flows towards the full result is tested. For this purpose we use the zero-dimensional theory, where the partition function (4) is a simple one-dimensional integral. This integral can be solved numerically, and in some limiting cases even analytically. For related works for real-valued effective actions and flows see [51–53]. We remark, that in $d = 0$ the partition function indeed develops zeros that are particularly difficult to resolve. In turn, in higher dimensions we expect cuts which may

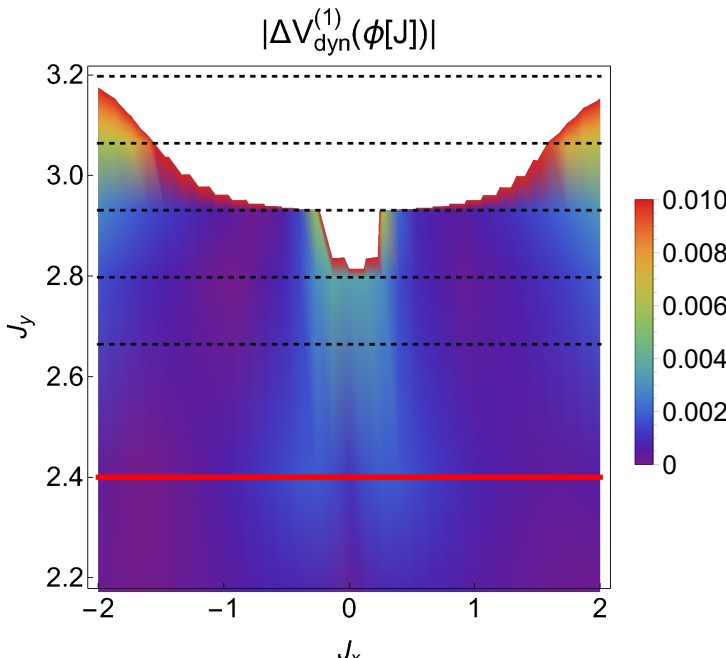

Figure 2: Absolute value of the numerical error of $V_{\text{dyn}}^{(1)}$ computed with the RG-adapted flow. The error is measured by subtraction from the numerically performed integral, the definition is given in (59). The pole position is inferred from the numerical evaluation of the integral and situated at at $J_y = 3$. The flow is solved on the slices along the black dashed lines, remaining values are interpolated. The red line is drawn in as a reference. It indicates the last convergent computation using the Polchinski flow, see Appendix G. All units are given in terms of the UV mass $m = 1$ at an initial cutoff $\Lambda = 5$.

facilitate the numerical treatment. This has been already observed within lattice simulations with complex Langevin dynamics [3].

The dynamical potential $V_{\text{dyn}}$ is computed using the flow of the RG-adapted scheme, the Polchinski flow and the 1PI flow, see Table 1 for a summary of relevant equations in the different schemes. Detailed derivations of the latter two flows are found in Appendix G and Appendix H respectively. Computations are, as discussed in Section 4.3, performed on slices of constant $\phi_y$, using 1d-numerical grid, ranging from $\phi_x \in [-3, 3]$. On each slice a grid of $K = 60$ cells is used with a polynomial of order $N = 2$ in each cell. Afterwards, results need to be mapped from the fields to the current $\phi \to J$. In the RG-adapted scheme, this is done via the definition of the current in (31a), the approximation of the propagator in (51) and the contained mass term $m_k^2$ (52). The mass term is computed from (53). This computation is explained further in the following Section 5.3, see also Figure 5.

Figure 1 shows results up to their maximal $J_y$ value, where the computation still converges. We find that the RG-adapted scheme retains a very high numerical accuracy in a very close proximity of the pole in comparison to the Polchinski flow and the 1PI flow. The convergence pattern is shown in Figure 2 and we find satisfying numerical accuracy of the RG-adapted scheme up to $J_y = 2.8$. The numerical error is given by the absolute value of $\Delta V_{\text{dyn}}^{(1)}(\phi[J])$ with,

$$\Delta V_{\text{dyn}}^{(1)}(\phi) = \Delta V_{\text{dyn,int}}^{(1)}(\phi) - \Delta V_{\text{dyn,flow}}^{(1)}(\phi). \tag{59}$$

Equation (59) is the difference of the numerical result for the first derivative of $V_{\text{dyn}}$ obtained

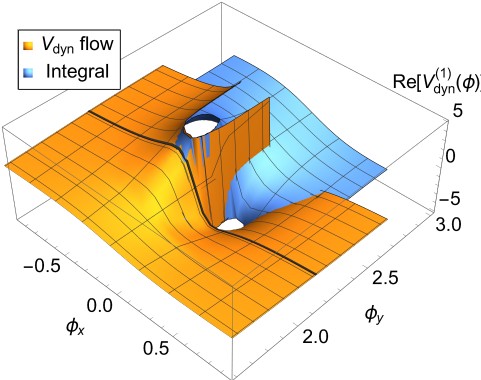

(a) Real part of $V_{\text{dyn}}^{(1)} = \partial_\phi V_{\text{dyn}}$ from the RG-adapted flow (57) in comparison to the exact numerical result.

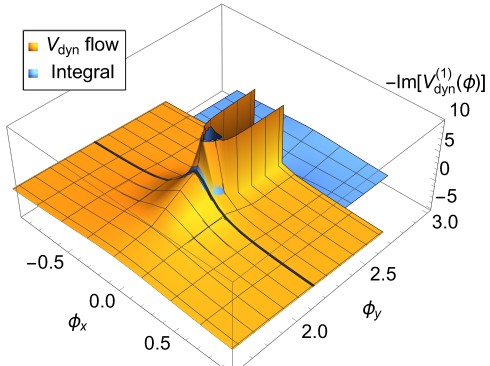

(b) The imaginary part of $-V_{\text{dyn}}^{(1)} = -\partial_\phi V_{\text{dyn}}$ from the RG-adapted flow (57) in comparison to the exact numerical result.

Figure 3: The plots show the first derivative of the dynamical potential $V_{\text{dyn}}^{(1)}(\phi)$ in $d = 0$ dimensions in the vicinity of the first Lee-Yang singularity. We compare results generated by the RG-adapted flow (56), (57) to the numerically integrated expression in (4). The numerical data has a slight offset which allows for a better graphical presentation. Both calculations are in agreement for $\phi_y$ smaller than the black line. The absolute difference between both results is shown in Figure 2. At the Lee-Yang singularity the RG-adapted flow freezes in and we see an un-physical continuation of the singularity. All units are given in terms of the UV mass $m = 1$ at an initial cutoff $\Lambda = 5$.

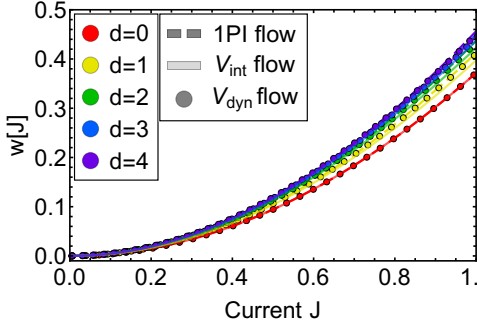

(a) Graphical comparison of the desity of the Schwinger functional $w[J]$ (61) as a function of the real current J.

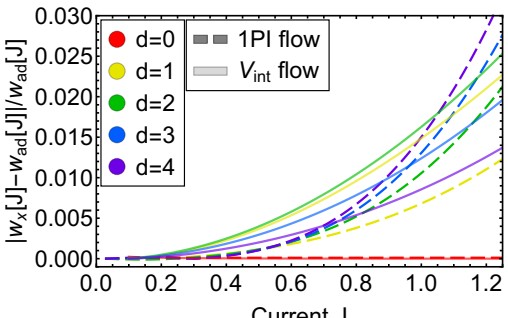

(b) The relative error of the density of the Schwinger functional $w_x$ in relation to the RG-adapted result $w_{\text{ad}}$. Where $x$ indicates the result from the Polchinski and 1PI flow as indicated by the legend.

Figure 4: Full potential of the Schwinger functional (61) on the real axis for different dimensions $d = 0, \ldots, 4$. We compare numerical results from the RG-adapted flow ($V_{\text{dyn}}$), the Polchinski flow ($V_{\text{int}}$) and the 1PI flow. The data was then used to compute the density of the Schwinger functional $w[J]$ using (61) and the information linked in Table 1. The results in $d = 0$ are exact, whereas the solutions for $d > 0$ start differing more strongly with increasing field values. All units are given in terms of the UV mass $m = 1$ at an initial cutoff $\Lambda = 5$.

from the direct numerical evaluation of the integral in (4), $V_{\text{dyn,int}}$, and the integration of the RG-adapted flow, $V_{\text{dyn,flow}}$.

From the numerical integration of (4) the pole position is found at

$$J = 3\mathrm{i}\,. \tag{60}$$

A computation of the Polchinski flow (G.2) on the same numerical grid, only converges up to $J_y = 2.4$, which is indicated by the red line in Figure 2. This significant increase in accuracy between the RG-adapted scheme and the Polchinski flow supports the expansion about $\phi_0$.

Within the current scheme, the 1PI flow is only convergent in the symmetric phase, i.e. for $\phi_y \le 1$, where $V_{\text{eff,k}}^{(2)} \ge 0$ with our initial conditions discussed in Section 5.1.2. A detailed discussion is given in Appendix H, an alternative formulation of the 1PI flow is subject to further investigation.

Beyond the Lee-Yang zero, an expansion about $\phi_0$ is no longer feasible. For all values $\phi_y > \phi_c$, where $\phi_c$ is the position of the Lee-Yang singularity, the flow runs into a *blow-up*, i.e. a singularity, compare to the raw data in Figure 3. A detailed discussion of the *blow-up* is given in Appendix F. The use of possible expansion points beyond the first pole are subject of ongoing investigations.

## 5.3 Convergence for $d \ge 0$ on the real axis

For higher dimensions, d> 0, the local potential approximation is indeed an approximation. Moreover, the LPA differs for the different schemes and we expect small deviations in the effective potentials.

Within a first application of the RG-adapted scheme to a quantum-field theory, i.e. $d > 0$, we investigate its behaviour on the real axis in comparison to established methods such as the Polchinski flow and the 1PI flow.

The raw LPA data in the RG-adapted scheme is converted by using the relations in Table 1. It provides us with the density of the Schwinger functional $w[J]$,

$$w[J] = W[J]/\mathcal{V}_d \quad \text{and} \quad \mathcal{V}_d = \int \mathrm{d}^d x\,. \tag{61a}$$

It follows,

$$
\begin{aligned}
w[J] &= \frac{1}{2}m_{k=0}^2\ \phi^2 - V_{\text{dyn}}(\phi)\big|_{\phi=G_k[\phi_0]J} \\
&= \frac{1}{2}\phi^2 - V_{\text{int}}(\phi)\big|_{\phi=\left(S^{(2)}\right)^{-1}J} \\
&= J\ \phi - V_{\text{eff}}(\phi)\big|_{\phi=\left(\frac{\partial V_{\text{eff}}}{\partial \phi}\right)^{-1}(J)}\,.
\end{aligned}
\tag{61b}
$$

The first, second and third line are given in terms of the RG-adapted result, the result of the Polchinski flow and the 1PI flow respectively. The different definitions of the current make use of the RG-adapted propagator $G_k[\phi_0]$ and the classical dispersion $S^{(2)}$. The coordinate change in the 1PI scheme follows directly from the Legendre transformation. The results from the different flows agree remarkably well, as is shown in Figure 4: the results agree necessarily in $d = 0$. In $d > 0$ we notice a deviation in the percent range for currents $J < 0.8$, which keeps increasing for even higher fields. Interestingly, we find that the results obtained from the 1PI flow, are in better agreement with that from the RG-adapted scheme for currents $J \lesssim 0.7$. This is likely related to the similar definition of RG-adapted currents, since in the 1PI case we have

$$J = \frac{\partial V_{\text{eff}}}{\partial \phi} + k^2\phi = (V_{\text{eff}}^{(2)} + k^2)\ \phi\,, \tag{62}$$

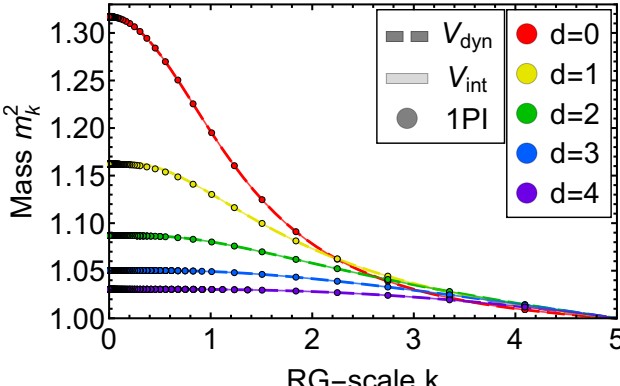

Figure 5: RG-scale dependence of the mass $m_k^2$ at the expansion point, as it is defined in (52), for dimensions $d = 0, \ldots, 4$. Results are obtained from computations of $V_{\text{dyn}}$ (RG-adapted scheme), $V_{\text{int}}$ (Polchinski flow) and the 1PI flow, see Table 1. All units are given in terms of the UV mass $m = 1$ at an initial cutoff $\Lambda = 5$.

due to the $Z_2$ symmetry of the potential. In this picture, the 1PI effective action is related to the RG-adaptation the current suggested in (31), using the full field-dependent propagator. Hence it is suggestive, that an RG-adapted scheme using the fixed field value $\phi_0$ compares well at small fields.

The behaviour shown in Figure 4 also suggests to compare the two-point function at the origin, the mass parameter $m_k^2$ (52) as it constitutes the defining difference in the presented schemes. In the RG-adapted scheme, it is computed in a separate equation from the potential using (53), whereas it can be read off of $V_{\text{int}}^{(2)}[0]$ (Polchinski) and $V_{\text{eff}}^{(2)}[0]$ (1PI). The results compare well for all dimension and are shown in Figure 5. This check shows that physical values can be extracted consistently from the various schemes in all dimensions.

## 6 Complex effective potential and Lee-Yang zeroes in $d = 4$

We now proceed with a comprehensive analysis of results for the complex effective potential and the properties of the Lee-Yang zeros in the $d = 4$ dimensional scalar field theory. As mentioned before, this theory or rather its O(4) variant is (part of the) scalar-pseudoscalar meson sector in the fRG approach to QCD with dynamical hadronisation, see [25, 60–62]. Roughly speaking, its embedding in QCD leads to additional driving forces in the present setup and hence this extension is covered by the general discussion of the types of differential equations in Section 4. Finally we are interested in such a QCD analysis at finite temperature and density. There the respective phase transitions and Lee-Yang zeros are related to the O(4) theory in $d = 3$ dimensions. We have performed such a numerical analysis also in $d = 3$ and $d = 1, 2$. However, a discussion of the location of the Lee-Yang zeros, their parameter dependence and the critical physics goes beyond the scope of the present paper, and will be presented elsewhere. Here we only state, that the present numerical analysis is also converging in $d = 1, 2, 3$.

In Section 6.1 we initiate the analysis with the discussion of the effective potential and its scaling properties. In Section 6.2 we evaluate the properties of the Lee-Yang singularity. In particular we compute the mass dependence of its location and the intersection point with the real axis at the phase transition point of the real scalar theory.

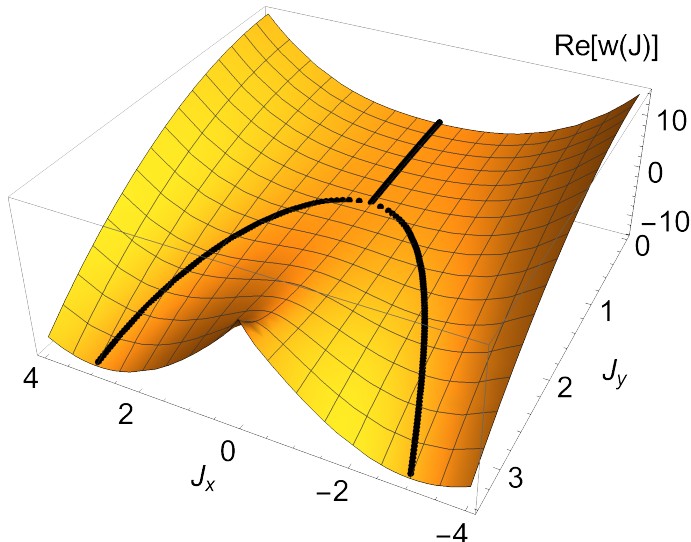

Figure 6: Real part of the density of the Schwinger functional $w[J]$. The data is obtained from the RG-adapted scheme. The raw data was computed in Section 6 and integrated using the path described in Appendix J and (61). The black dotted line indicates the expectation value for the magnetisation, compare (63), and is evaluated for all $\phi_y$ using (63b) with $H = 0$. The critical value for the spontaneous breaking of symmetry is clearly visible at $J_0 \approx 1.37$. All units are given in terms of the UV mass $m = 1$ at an initial cutoff $\Lambda = 5$.

## 6.1 Scaling properties and Lee-Yang singularities

For the evaluation of the scaling properties in the vicinity of the Lee-Yang singularities we evaluate the full effective potential in the complex plane. The computations are performed on $\phi$ slices with constant imaginary part $\phi_y$ with a spacing of 0.1 until $\phi_y = 3.2$. At $\phi_y = 3.2$ we are in the proximity of the *blow-up*, investigated in Appendix F. Furthermore, we increase the resolution in the critical area between $\phi_y \in [1.3, 1.4]$ to 0.01. The slices are interpolated in $\phi_y$ direction after the evaluation. Lastly, the effective potential is computed from the raw data, following the integration procedure described in Appendix J.

Performing calculations in the complex plane is tantamount to scanning the theory for different initial masses, since $m^2[0, \phi_y] = m^2 - \frac{\phi_y^2}{2}$ on each slice. In $d = 4$, the $\phi^4$ theory is evaluated in its critical dimension, and hence we expect mean field scaling at the Lee-Yang singularity.

In the present case, mean field scaling occurs in a relatively big scaling regime, which allows for very accurate fits. Once the scaling amplitudes are determined, the location of the Lee-Yang singularity is estimated in Section 6.2. The order parameter is given by the average magnetisation $M$ which is the solution of the equation of motion (EoM),

$$M(\phi_y, H) = \phi_{\text{EoM}}. \tag{63a}$$

The solution to the EoM, $\phi_{\text{EoM}}$ is determined from

$$\left. \frac{\partial w[J(\phi_x, \phi_y)]}{\partial \phi_x} \right|_{\phi_x = \phi_{x,\text{EoM}}} = H. \tag{63b}$$

The solution $\phi_{\text{EoM}}$ in (63) depends on the effective mass $m^2[0, \phi_y]$ of the corresponding $\phi_y$-slice and the external field $H$. The magnetisation (63) follows the minimum of the density

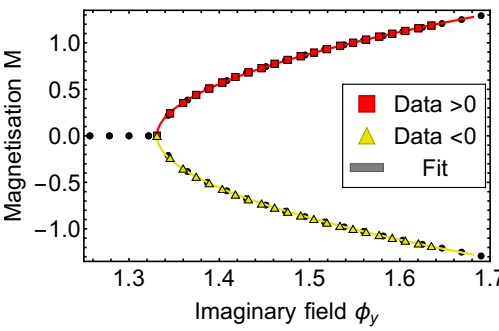

(a) Real part of the magnetisation as a function of $\phi_y = \mathrm{Im}\,\phi$.

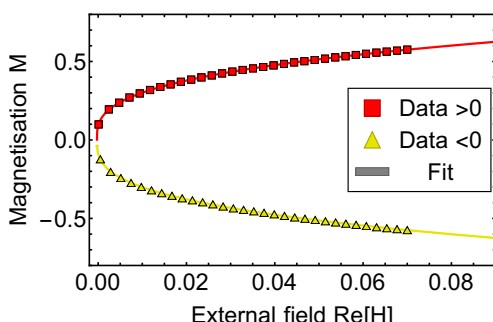

(b) Real part of the magnetisation as a function of $\mathrm{Re}\,H$.

Figure 7: Real part of the average magnetisation $M$ as a function of the imaginary part $\phi_y$ of the field (left) and the real part $\mathrm{Re}\,H$ of the external field (right). The fits are based on the scaling functions (64) with the fit parameters in Table 2. Both branches of the magnetisation are fitted separately, and the overall error of the fit parameters is obtained by averaging over both branches and taking the fit error into account. All units are given in terms of the UV mass $m = 1$ at an initial cutoff $\Lambda = 5$.

$w$ and is depicted in Figure 6. There, the critical value $J_0 \approx 1.37$ is clearly visible as the bifurcation point of the minimum.

For the extraction of the scaling exponents we use the scaling relations in the vicinity of the critical $\phi_y$-slice

$$\mathrm{Re}\left[M(\phi_y, H = 0)\right] = B\left(\frac{\phi_y - \phi_c}{\phi_c}\right)^{\beta},$$

$$\mathrm{Re}\left[M(\phi_y = \phi_c, H)\right] = B_c H^{1/\delta}, \tag{64}$$

where $\phi_c$ is the critical field that signals the onset of symmetry breaking. The amplitudes $B, B_c$ are used later for computing universal quantities.

A $\chi$-squared fit is performed separately on both branches and the difference between fits is used for an error estimate of the numerical error.

The fit parameters are given in Table 2 and the result is plotted in Figure 7. From the scaling fits we obtain

$$\beta = 0.505(23), \qquad \delta = 2.992(18), \tag{65}$$

which corresponds well to the expected mean field scaling parameters $1/2$ and $3$ respectively. The error is estimated by dropping the last five data-points. Since we do not aim at a precise estimate of the scaling parameters in this work, we refrain from an in-depth error analysis. Possible sources of error are the interpolation of data in $\phi_y$-direction, as well as the determination of the minimum.

A further, derived, scaling exponent is given by,

$$\gamma = \beta(\delta - 1), \tag{66}$$

and governs the scaling of the susceptibility $\chi$.

$$\chi(\phi_y, H = 0) = \frac{\partial M}{\partial H}(\phi_y, H)\big|_{H=0}. \tag{67}$$

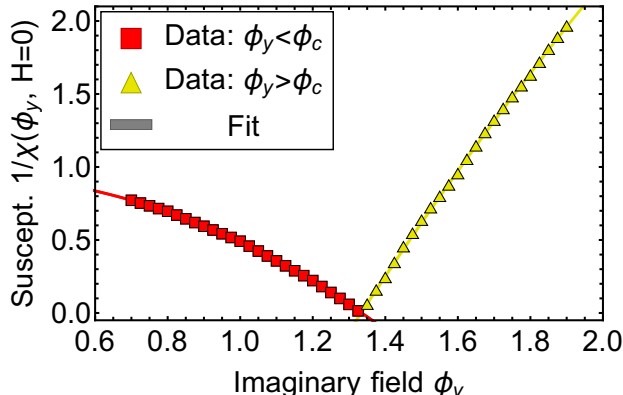

Figure 8: Inverse of the susceptibility $\chi$ as a function of field $\phi_y = \text{Im}\,\phi$. Fits to the data are performed with (68), where we use $\gamma = 1$. The resulting scaling amplitudes are given in (69). Small wiggles in the data-points originate from an 2D-interpolation of the raw data. All units are given in terms of the UV mass $m = 1$ at an initial cutoff $\Lambda = 5$.

We include an additional subleading quadratic term in our fit to allow for a bigger fit-interval and a better fit to the data. The fit function reads

$$\chi(\phi_y, H = 0) = C_\pm \left( \frac{\phi_y - \phi_c}{\phi_c} + D \left( \frac{\phi_y - \phi_c}{\phi_c} \right)^2 \right)^{-\gamma}, \tag{68}$$

where $\gamma = \beta(\delta - 1) = 1$ is the mean field scaling exponent. $C_\pm$ are the scaling amplitudes in the symmetric and spontaneously broken phase respectively. The susceptibility is obtained from $\partial_{\phi_x}^2 V_{\text{dyn}}(\phi_x, \phi_y)$, for details see Appendix I.

We have applied a $\chi$-squared fit to the inverse of the susceptibility $\chi(\phi_y, H = 0)^{-1}$. The fit with (68) and the data-points are shown in Figure 8. The overall amplitudes $C_\pm$ and the relative factor $D$ of the subleading term from the fits are given by

$$\begin{aligned} C_+ &= -0.4369(11), \quad C_- = 0.2025(21), \\ D_+ &= -0.256(20), \quad D_- = 0.6117(88). \end{aligned} \tag{69}$$

In mean field, the amplitudes $C_\pm$ are related with $|C_+| = 2|C_-|$. This relation is violated by approximately 8%. The deviation in $C_\pm$ also persists for smaller fit-intervals. The numerical errors may be caused by a small true scaling regime as well as the interpolation of $\phi_y$ slices. This behaviour may also be linked to a faulty smoothing out of an cut for $\phi_y > \phi_c$. All these potential sources are currently investigated.

Gathering all scaling parameters, we obtain the universal scaling amplitude

$$R_\chi = \frac{C_+ B^{\delta-1}}{B_c^\delta} = 1.035(25). \tag{70}$$

This agrees well with the expected value in mean field computations $R_\chi = 1$. Computing $R_\chi$ from $C_-$ yields $R_\chi = 0.931(22)$. This result also suggests, that the potential smoothing out of the cut does not strongly affect the solution of the equations of motion. The fit result also provides an error for the critical field with

$$\phi_c = 1.3340(10), \tag{71}$$

where we have taken the mean of both fits (69) and their respective error.

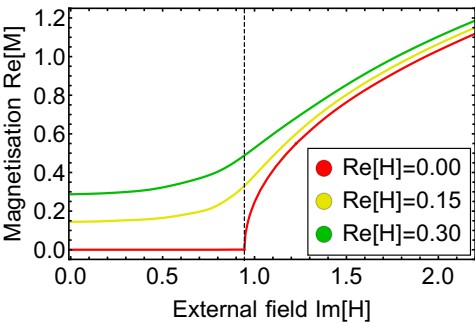

(a) Real part of the magnetisation.

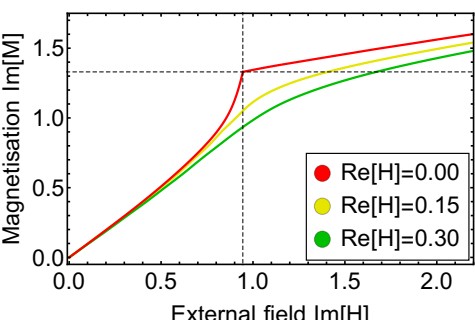

(b) Imaginary part of the magnetisation.

Figure 9: Real and imaginary part of the average magnetisation $M$, defined in (63), at $\phi_y = \phi_c$ as a function of the imaginary part $\operatorname{Im} H$ of the external field. The magnetisation is shown for different values of the real part $\operatorname{Re} H = 0$, 0.15, 0.30 of the external field. The dashed lines highlight the position of the cusp at $M = 1.33i$ and $\operatorname{Im} H = 0.9448$. The Lee-Yang singularity is clearly visible at $\operatorname{Re} H = 0$ and is increasingly smudged out for higher values of the real external field. All units are given in terms of the UV mass $m = 1$ at an initial cutoff $\Lambda = 5$.

Table 2: Fit parameters for the fit (64) applied to the data-set in Figure 7. The error is the $\chi^2$ error of the fit. We keep fits of both branches to get an estimate for the error of the overall scaling. The error on $T_c$ is $\leq 10^{-5}$ and not indicated.

|  | $\beta$ | $\delta$ | $\lvert B \rvert$ | $\lvert B_c \rvert$ | $\phi_0$ |
|---|---|---|---|---|---|
| $\phi_{x,\mathrm{EoM}} > 0$ | 0.5050(40) | 2.9830(88) | 2.520(20) | 1.408(40) | 1.331 |
| $\phi_{x,\mathrm{EoM}} < 0$ | 0.5060(43) | 3.000(11) | 2.526(22) | 1.3933(55) | 1.331 |

## 6.2 The Lee-Yang Singularity

Finally, we discuss the location of the Lee-Yang edge singularity. This is chiefly important for an application of the current approach to QCD: the extrapolation of the location of the Lee-Yang singularity in QCD as a function of baryon chemical potential onto the real axis constrains the location of a potential critical end point. More generally it constrains the location of the onset of new physics.

In the present $\phi^4$ model case, the edge singularity in $d = 4$ corresponds to a kink in the magnetisation. As an example, we use the data-set from the previous Section with the initial conditions from Section 5.1.2 with a UV-mass $m^2 = 1$. We evaluate the magnetisation $M$ defined in (63) at critical $\phi_y = \phi_c$ and imaginary external field $H$. Both the real and imaginary part of the magnetisation are shown in Figure 9. The exact position of the Lee-Yang edge singularity is located at $M_{\mathrm{crit}} = 1.3320(20)i$ and $\operatorname{Im} H = 0.9448$. It can be identified by the second order phase transition in the real part $\operatorname{Re} M$, as well as the cusp in the imaginary part $\operatorname{Im} M$.

We have done a similar analysis in $d = 3$, where we find the position of the Lee-Yang edge singularity at $M_{\mathrm{crit}} = 1.3136(31)i$ and $\operatorname{Im} H = 0.9589$ for an UV-mass $m^2 = 1$. For a comprehensive analysis of the theory in $d = 3$, including also a discussion of the size of the (small) scaling regime is deferred to a work in preparation.

More generally, for initial masses $m^2 > m_c^2$ and fixed initial self coupling $\lambda = 1$, the expec-

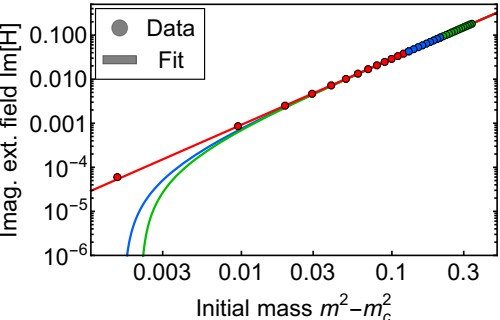

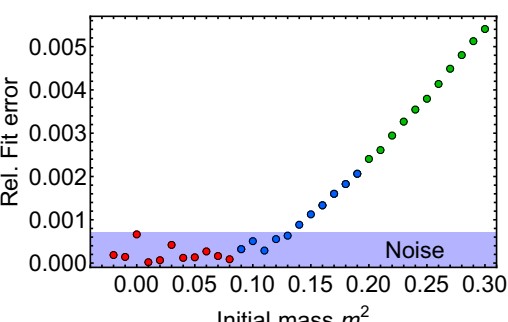

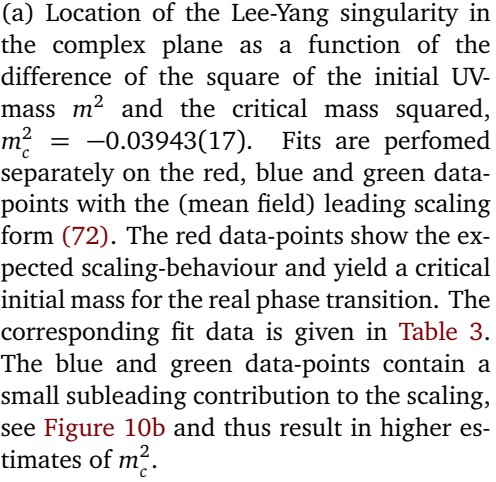

(a) Location of the Lee-Yang singularity in the complex plane as a function of the difference of the square of the initial UV-mass $m^2$ and the critical mass squared, $m_c^2 = -0.03943(17)$. Fits are perfomed separately on the red, blue and green data-points with the (mean field) leading scaling form (72). The red data-points show the expected scaling-behaviour and yield a critical initial mass for the real phase transition. The corresponding fit data is given in Table 3. The blue and green data-points contain a small subleading contribution to the scaling, see Figure 10b and thus result in higher estimates of $m_c^2$.

(b) Relative error between data-points and the fit obtained close to $m_c$: red fit-function in Figure 10a with the parameters Table 3. The first few data-points are exempt from the plot, since their absolute value is smaller than the fit-accuracy. The red, blue and green color corresponds to that of the data-points in Figure 10a. The noise is linked to a numerical error on the data, whose absolute value is smaller than $10^{-5}$. We find a subleading behaviour to the scaling for initial masses $m^2 \gtrsim 0.14$.

Figure 10: Location of the phase transition on the real axis from an extrapolation of the location of the Lee-Yang zeros. All computations use an initial cutoff $\Lambda = 5$.

tation value of the real field $\phi_x$ vanishes, $\phi_x(m^2 > m_c^2) = 0$. Accordingly, the $Z_2$ symmetry is preserved on the real axis. In turn, for smaller $m^2$, the $Z_2$ symmetry is spontaneously broken, $\phi_x(m^2 < m_c^2) \neq 0$. The critical value $m_c^2$ determines the location of the second order phase transition.

This allows us to determine the $m$-dependence of the critical field value $\phi_c(m)$, and in particular its endpoint with $\phi_c(m_c) = 0$. The result is shown in Figure 10a, were we show the value of the imaginary part of the magnetisation or solution of the equation of motion, $\phi_y = \operatorname{Im} M$, at the Lee-Yang singularity as a function of the initial UV-mass squared. We extract the critical UV-mass $m_c^2$ by fitting the universal scaling behaviour,

$$\operatorname{Im} H = A(m^2 - m_c^2)^\Delta. \tag{72}$$

The scaling fit with the fit parameters in Table 3 is based on the data-points with initial masses $m_c^2 < m^2 < 0.1$. We recover a critical UV-mass $m_c^2 = -0.03943(17)$. For all UV-input masses higher than $m_c^2$, we are in the symmetric phase, which can be seen from the IR-mass on the real axis. For example, performing a computation at a UV-mass of $m^2 = -0.039$, yields an IR-mass $m_0^2 = 4.2 \, 10^{-4}$. We also recover the mean-field scaling $\Delta = \beta\delta = 3/2$ remarkably well with our estimate given by $\Delta = 1.4969(66)$.

Finally, we investigate the size of the scaling regime. Scaling fits to data-points from initial masses $m^2 > 0.09$ yield a higher estimate for $m_c^2$, which hints at a small subleading contribution to the scaling behaviour (blue and green curves in Figure 10a). The relative error between the scaling-fit and the data-points is depicted in Figure 10b. We find a subleading contribution to the scaling for $m^2 > 0.14$ which grows in importance for higher initial masses $m^2$. At $m^2 = 1$ the relative fit error on the Lee-Yang location is only $\approx 3\%$. Hence, we deduce a large scaling regime in $d = 4$.

Table 3: Fit parameters to the scaling fit of the Lee-Yang location in (72) and plotted with the data-set in Figure 10a. The error to the fit is obtained by removing the last data point.

| $\Delta$ | $m_c^2$ | $A$ |
|---|---|---|
| 1.4969(66) | −0.03943(17) | 0.913(12) |

## 7 Summary and Outlook

In the present work we have set up general fRG approaches for computing complex actions. To that end we have compared different general fRG flows for the Wilsonian effective action and the 1PI effective action, based on the general flows for both, (12) and (27). The analysis suggests that the construction of adapted fRGs is key to constructing systems of partial differential equations whose types support flows towards the infrared, see Section 4.

The numerical results for Lee-Yang zeroes have been obtained from RG-adapted flows of the Wilsonian effective action. In turn, the naive implementation of the commonly used 1PI flow has a much smaller convergence radius in the complex plane. In our opinion, a fully RG-adapted 1PI flow based on (27) as well as a careful choice of the initial condition should resolve this issue, and we hope to report on this matter in the near future.

The present conceptual results allow for a systematic construction of these flows within the theory and truncation at hand. Our explicit numerical computations of the effective potential in scalar theories in zero to four dimensions have aimed at resolving Lee-Yang zeroes, and are based on the RG-adapted flow for the Wilsonian effective action constructed here. The setup allowed us to compute the full effective potential in the complex magnetisation plane, see in particular Figure 6. These results give access to the position as well as the mass-dependence of the Lee-Yang zeroes, see Figure 10a.

This evaluation of the mass dependence allowed us to predict the location $m_c^2$ of the phase transition from the symmetric into the broken phase in terms of an extrapolation of the Lee-Yang singularity and its intersection point with the real axis. It can be seen as a precursor of a respective computation in QCD, where the present scalar potential or rather scalar sector is linked to the scalar-pseudoscalar meson sector obtained via dynamical hadronisation in functional QCD. Then, the dependence of the location of the Lee-Yang zero on the chemical potential can be used to constrain the location of the critical end point or the onset of new physics within first principle QCD. We hope to report on the respective investigation in the near future.

## Acknowledgments

We thank J. Horak, F. Rennecke, M. Salmhofer, F. Sattler, V. Skokov, J. Urban and N. Wink for discussions.

**Funding information** This work is done within the fQCD collaboration [63], and is supported by the Studienstiftung des Deutschen Volkes and EMMI. This work is funded by the Deutsche Forschungsgemeinschaft (DFG, German Research Foundation) under Germany's Excellence Strategy EXC 2181/1 - 390900948 (the Heidelberg STRUCTURES Excellence Cluster) and the Collaborative Research Centre SFB 1225 - 273811115 (ISOQUANT).

# A The Polchinski flow

In this Appendix we briefly recapitulate the derivation of the Polchinski equation [14] for a scalar theory. All correlation functions in Euclidean field theory can be obtained from the generating functional. The generating functional is defined by its derivatives, see Equation (3). It can, however, also be linked to an explicit path integral representation;

$$Z_k[J] = \int d\varphi \, e^{-S[\varphi] - \frac{1}{2} \int_x \varphi R_k \varphi + \int_x J(x)\varphi(x)}, \tag{A.73}$$

for a given theory of the real scalar field, $\varphi \in \mathbb{R}$. In accordance with the general procedure of the functional renormalisation group, we have already introduced the (infrared) cutoff term in (A.73). The cutoff term $R_k$ suppresses all contributions with $p^2 < k^2$ to the generating functional. The correlation functions, derived from the generating functional $Z[J]$, are the full ones including their disconnected parts. The connected parts are derived from the Schwinger functional

$$W_k[J] = \log Z[J]. \tag{A.74}$$

Important examples are given by the mean field in a given background current,

$$\phi[J] = \frac{\delta W_k[J]}{\delta J}, \tag{A.75}$$

and the propagator,

$$G_k = \langle \varphi(x)\varphi(y) \rangle_c = W_k^{(2)}[J], \tag{A.76}$$

where the subscript $_c$ stands for *connected*. The flow of $W_k[J]$ is given by

$$\partial_t W_k[J] = -\frac{1}{2} \mathrm{Tr} \, \partial_t R_k \left[ W^{(2)}[J] + \left( W^{(1)}[J] \right)^2 \right]. \tag{A.77}$$

Equation (A.77) and its generalisations are the master equations for the derivation of flow equations for the Wilsonian effective action (generating functional of *amputated connected* correlation functions), the 1PI effective action (generating functional of *one particle irreducible* correlation functions), functional symmetry identities, and further generating functions.

The derivation of the Wilsonian effective action continues, by using the inverse classical propagator in the current. This removes (amputates) the external legs from the Schwinger functional $W_k[J]$,

$$J = S_k^{(2)} \phi, \quad \text{with} \quad S_k^{(2)} = S^{(2)}[\phi_0] + R_k, \tag{A.78}$$

with a given background $\phi_0$, which can be chosen conveniently. The respective generating functional,

$$S_{\mathrm{eff},k}[\phi] = -W_k[S_k^{(2)}\phi], \tag{A.79}$$

is the generating functional of *amputated connected* correlation functions. This amputation is elucidated at the example of the one- and two-point functions,

$$S_k^{(2)} \frac{\delta W_k[J]}{\delta J} = -\frac{\delta S_{\mathrm{eff},k}[\phi]}{\delta \phi},$$

$$S_k^{(2)} \frac{\delta^2 W_k[J]}{\delta J^2} S_k^{(2)} = -\frac{\delta^2 S_{\mathrm{eff},k}[\phi]}{\delta \phi^2}. \tag{A.80}$$

The flow equation for the Wilsonian effective action $S_{\text{eff},k}[\phi]$ can be obtained by inserting (A.79) or rather (A.80) into the master-equation (A.77). It is given by

$$\left(\partial_t + \int_x \phi\, S_k^{(2)}\, \partial_t G_k^{(0)}\, \frac{\delta}{\delta\phi}\right) S_{\text{eff},k}[\phi] = \frac{1}{2}\,\text{Tr}\,\partial_t G_k^{(0)}\left[S_{\text{eff},k}^{(2)}[\phi] - \left(S_{\text{eff},k}^{(1)}[\phi]\right)^2\right], \qquad (A.81)$$

with the classical propagator

$$G_k^{(0)} = \frac{1}{S_k^{(2)}} = \frac{1}{S^{(2)}[\phi_0] + R_k}. \qquad (A.82)$$

This is Wegner's flow (12) with the kernel (19) and anomalous dimension (20) as discussed in Section 2.1.1.

Note that for cutoff-dependent (evolving) backgrounds $\phi_0$ the $t$-derivative in $\partial_t S_k^{(2)}$ also hits the field. When expanding the Wilsonian effective action about its classical (or rather UV) counter part $S_k[\phi]$, the trivial flow for the cutoff term is apparent. Already in the case of real external fields, we find some inconveniences with this formulation. For example in the investigation of chiral symmetry breaking, $G_k^{(0)}$ runs into a singularity at some $k > 0$, thus necessitating a formulation in evolving backgrounds. This observation motivates the choice of an expansion about an RG-adapted propagator (see Section 3), already in a real setting.

Continuing with the derivation of the classical expansion of the Polchinski flow, we separate the full two-point function from the effective action,

$$S_{\text{eff},k}[\phi] = S_{\text{int},k}[\phi, \phi_0] - \frac{1}{2}\int_x \phi\, S_k^{(2)}[\phi_0]\, \phi. \qquad (A.83)$$

This split eliminates the trivial running of $S^{(2)}[\phi_0]$ from the flow and makes numerical computations more convenient. Inserting (A.83) into the Polchinski flow (A.81) leads us to the flow of the interaction part $S_{\text{int},k}[\phi]$

$$\partial_t S_{\text{int},k}[\phi] = \frac{1}{2}\,\text{Tr}\,\partial_t G_k^{(0)}\left[S_{\text{int},k}^{(2)}[\phi] - \left(S_{\text{int},k}^{(1)}[\phi]\right)^2\right]$$

$$- \frac{1}{2}\,\partial_t G_k^{(0)}\, S_k^{(2)}, \qquad (A.84)$$

where the second line is $\phi$-independent, but $\phi_0$-dependent.

# B  Field expansion and flows of $n$-point functions of the RG-adapted flow

This Appendix contains some technical details of the derivation of the RG-adapted flow derived in Section 3.1. With (32), the one-point function is simply

$$\bar{\phi} = -G_k[\phi_0]\, S_{\text{ad},k}^{(1)}[0], \quad \text{with} \quad \bar{\phi} = \langle\varphi\rangle_{J=0}. \qquad (B.1)$$

This entails that the one-point function encodes the information about the expectation value $\bar{\phi}$ of the field (up to the propagator). The latter is given by the two-point function,

$$S_{\text{ad}}^{(2)}[0, \phi_0] = -G_k^{-1}[\phi_0] = -\left(\Gamma_k^{(2}[\phi_0] + R_k\right), \qquad (B.2)$$

with the 1PI effective action $\Gamma_k[\phi]$. We emphasise that (B.2) entails that the argument $\phi$ of $S_{\text{ad},k}^{(2)}$ is the difference field to $\phi_0$, the possibly $k$-dependent expansion point. We have

$$S_{\text{ad}}[\phi] = -\frac{1}{2}\int (\phi + \bar{\phi})G_k^{-1}[\phi_0](\phi + \bar{\phi}) + \Delta S_{\text{eff}}[\phi, \bar{\phi}]. \tag{B.3}$$

Finally, the higher $n$-point functions $S_{\text{eff},k}^{(n>2)}$ encode the interactions. We now disentangle the flow of the latter from that of the propagator $G_k[\phi_0]$ or rather $\Gamma_k^{(2)}[\phi_0]$. This is the crucial ingredient of the RG-adapted flow for $S_{\text{dyn}}$, (39a) in Section 3.1, and reads

$$\partial_t \Gamma_k^{(2)}[\phi_0] = \frac{1}{2}\operatorname{Tr} \mathcal{C}_k \left[ S_{\text{dyn},k}^{(4)}[0] - 2S_{\text{dyn},k}^{(1)}[0]S_{\text{dyn},k}^{(3)}[0]\right]. \tag{B.4}$$

In (B.4) the vertices $S_{\text{dyn},k}^{(3,4)}[0]$ enter as well as the one-point function. The latter is given by (B.1) with the flow

$$\left(\partial_t + \gamma_{\text{dyn},k}\right)S_{\text{dyn},k}^{(1)}[0] = \frac{1}{2}\operatorname{Tr} \mathcal{C}_k S_{\text{dyn},k}^{(3)}[0]. \tag{B.5}$$

Finally, the flow of the interaction part,

$$S_{\text{int},k}[\phi] = S_{\text{dyn},k}[\phi] - S_{\text{dyn},k}^{(1)}[0]\phi, \tag{B.6}$$

is given by

$$\left(\partial_t + \int_x \left[\phi\, \gamma_{\text{dyn},k} + \mathcal{D}_k\right]\frac{\delta}{\delta\phi}\right)S_{\text{int},k}[\phi] = \frac{1}{2}\operatorname{Tr}\mathcal{C}_k\left[\hat{S}_{\text{int},k}^{(2)}[\phi] - \left(S_{\text{int},k}^{(1)}[\phi]\right)^2\right], \tag{B.7}$$

with

$$\mathcal{D}_k = \mathcal{C}_k S_{\text{dyn},k}^{(1)}[0],$$

$$\hat{S}_{\text{int},k}^{(2)}[\phi] = S_{\text{int},k}^{(2)}[\phi] - S_{\text{int},k}^{(3)}[0]\cdot\phi - \frac{1}{2}S_{\text{int},k}^{(4)}[0]\cdot\phi^2. \tag{B.8}$$

The definition of the Schwinger functional in the complex plane suggests an ambiguity in the definition of the complex part of (B.7) due to the complex logarithm. This problem is discussed in Appendix D.

## C  Large cutoff limit

To begin with, one can easily convince oneself that for $R_k \to \infty$ in the limit $k \to \infty$, where the path integral gets approximately Gaußian, to wit

$$S_{\text{eff},k}[\phi] \overset{k\to\infty}{\longrightarrow} -\frac{1}{2}\int_x \phi\left(S_k^{(2)} + R_k\right)\phi + O(\phi^3). \tag{C.1}$$

The Wilsonian effective action tends towards the classical Wilsonian action, including the cutoff term, in the UV. This property is essential in deriving the initial conditions. Strictly speaking it tends towards the UV-relevant part of the Wilsonian effective action. This holds true for sufficiently small fields.

The case of general fields is resolved in a indirect way. We utilise that the flow of the effective action $\Gamma_k$ decays for large fields and only the primitively divergent terms in the action flow. This leaves us with the limit

$$\Gamma_{k\to\infty}[\bar{\phi}] \to S_{\mathrm{cl}}[\bar{\phi}]. \tag{C.2}$$

This implies that the relation between the current and the (mean 1PI) field $\bar{\phi}$ is given by

$$J = \frac{\delta\Gamma_k}{\delta\bar{\phi}} + R_k\bar{\phi}. \tag{C.3}$$

For the classical action

$$S_{\mathrm{cl}}[\bar{\phi}] = \frac{1}{2}\int \bar{\phi}S^{(2)}\bar{\phi} + \frac{\lambda}{4!}\bar{\phi}^4, \tag{C.4}$$

we arrive at

$$J = \left(S^{(2)} + R_k\right)\bar{\phi} + \frac{\lambda}{6}\bar{\phi}^3, \tag{C.5}$$

which entails that the 1PI mean field $\bar{\phi}$ and the Wilsonian field $\phi$ in (A.78) agree up to the interaction piece. We have

$$\phi = \bar{\phi} + \frac{\lambda}{6}G_k^{(0)}\bar{\phi}^3. \tag{C.6}$$

Evidently, for sufficiently small field we have $\phi \approx \bar{\phi}$ and the Wilsonian action tends (C.1). In turn, for large fields $\phi \to \infty$ we have

$$\phi \approx \frac{\lambda}{6}G_k^{(0)}\bar{\phi}^3 \quad \longrightarrow \quad \bar{\phi} \approx \left(\frac{6}{\lambda}S_k^{(2)}\phi\right)^{\frac{1}{3}}. \tag{C.7}$$

In any case we have for $k \to \infty$,

$$W_k[J] \to \frac{1}{2}\int \bar{\phi}S_k^{(2)}\bar{\phi} + \frac{\lambda}{8}\int \bar{\phi}^4, \tag{C.8}$$

and hence

$$S_{\mathrm{eff},k}[\phi] \to -\frac{1}{2}\int \bar{\phi}S_k^{(2)}\bar{\phi} - \frac{\lambda}{8}\int \bar{\phi}^4, \tag{C.9}$$

where $\bar{\phi}[\phi]$ solves (C.6).

# D  Cuts

In this section we are going discuss why the branch cuts of the complex logarithm do not factor into the computation. The Schwinger functional is defined as the logarithm of the generating functional (A.73). In a complex formulation we have,

$$W_k[J] = \log\left(|Z_k[J]|\right) + i\,\mathrm{Arg}(Z_k). \tag{D.1}$$

Just from the analytic expression we expect the branch-cuts of the logarithm to factor into the computation. There are now several options to deal with the complex part of the logarithm:

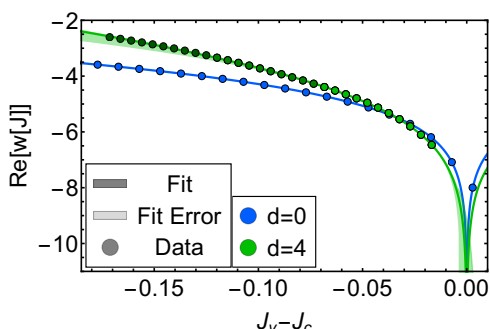

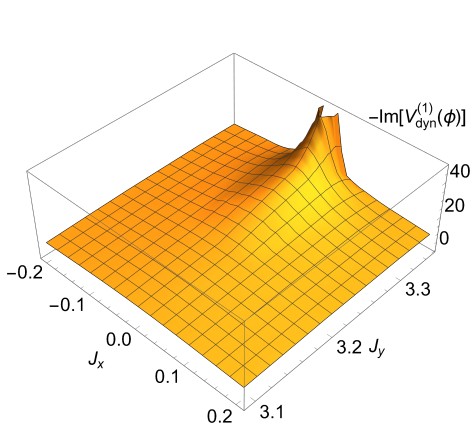

(a) Raw data: Negative imaginary part of the first derivative of the effective interaction potential $-\mathrm{Im}[\partial_{\phi_x} V_{\mathrm{dyn,k}}(\phi)]$. Computations converged up to $J_y = 3.35$.

(b) Fit of the position of the Lee-Yang singularity on data points of the density $\mathrm{Re}[w(J)]$. The singularity in the $d = 0$ case is fitted as a consistency check of the fit function. The fit error band is obtained by adding/removing 10 data points from the fitted dataset. The data points in question are shaded darker. The original fit is performed on the two lighter shades.

Figure 11: Plots for the computations on the position of the numerical blow-up. The blow-ups are located at $J_c = 3.3659(42)$i (and $J_c = 3.002(1)$i) in $d = 4$ ( and $d = 0$). We show the raw data in $d = 4$, which is directly obtained from the RG-adapted flow (56). From this we compute the full potential (61) and fit the exact position of the singularity. The fit-function is indicated in (F.2) and the fitted parameters are given in Table 4. All units are given in terms of the UV mass $m = 1$ with an initial cutoff $\Lambda = 5$.

- The logarithm is defined such that the Schwinger functional remains continuous. This is possible by remaining on the same sheet.

- The logarithm is defined as per usual on an interval $[-\pi, \pi]$. However (A.81) implies that the flow is only dependent on the first derivative of $W_k$. The branch cuts in (D.1) therefore only appear in the equation on a set with volume zero and do not contribute to the flow.

For the purpose of this paper we chose the first option. This options allows a trivial expansion of the initial conditions in Appendix C to the complex plane.

# E  Numerical evaluation of higher derivatives

In this section we explain how to extract quantitatively precise n-point vertices from the full, field-dependent potentials. For this purpose we need to take a closer look at the (L)DG-method, which is used to solve the field-dependent flows, see Table 1 for an overview of the flow equations. The Discontinuous Galerkin method (DGM) was originally developed for simple conservation laws of purely convective nature, that is equations which do not contain any higher derivative operators, such as for example diffusion terms. The main idea of the DGM is to combine the discontinuous nature and geometric flexibility of Finite Volume methods (FVM) with the higher order accuracy of Finite Element methods (FEM). Hence the computational

domain $\Omega_h$ is split up into $K$ non-overlapping elements $D^k$ (FVM features)

$$\Omega \simeq \Omega_h = \bigcup_{k=1}^{K} D^k \,, \tag{E.1}$$

and each element $D^k$ uses a polynomial of order $N$ to approximate the numerical solution $u_h$

$$u_h^k(t,x) = \sum_{n=1}^{N+1} \hat{u}_n^k(t)\psi_n(x)\,, \tag{E.2}$$

where the polynomial basis $\{\psi_n\}$ is given by the Legendre-polynomials with the corresponding coefficient $\hat{u}_n^k$ (FEM features). To ensure convergence across element-interfaces, a so called numerical flux must be introduced at the borders of each element. For a more throughout discussion see [38].

In this paper, higher order derivatives are used within the numerical computation in two different ways.

- Diffusive terms are contained in the flows, i.e. field-dependent second order derivatives of the solution itself. These terms require non-trivial additional numerical fluxes, which are provided in the LDG scheme. The scheme is an extension of the simple DG method and briefly introduced in Section 4.2, for a detailed discussion of the LDG method within an fRG context see [46].

- Higher order derivatives of the solution, evaluated at an expansion point $x_0$. These terms are directly evaluated from the solution by taking derivatives of the functional basis

$$\partial_x u_h^k(t,x)|_{x=x_0} = \sum_{n=1}^{N+1} u_n^k(t)\partial_x \psi_n(x)|_{x=x_0}\,, \tag{E.3}$$

  and feed back into the flow in a trivial manner. All computations generating data at the expansion point are therefore preformed using a polynomial order $N > 4$.

For specific, strongly dynamical scenarios, the second procedure may lead to apparent convergence towards a false solution. In the current computation we test convergence of the derivatives by making use of some known symmetry properties. The $Z_2$ symmetry of the potential $V$ requires $\partial_x^n V|_{x=x_0} = 0$ for odd $n$, thus we track the numerical values of $\partial_x V|_{x=x_0}$ and $\partial_x^3 V|_{x=x_0}$ throughout the $d = 0$ computation, which displays the highest dynamics and generates the highest numerical error for this check. We find that

$$1.02476e-13 \leq |\partial_x V|_{x=x_0}| \leq 2.18304e-12\,,$$
$$2.7792e-10 \leq |\partial_x^3 V|_{x=x_0}| \leq 9.13647e-09\,, \tag{E.4}$$

for a polynomial order $N = 6$.

# F  The parabolic blow-up

In Section 5.2 and Section 6 we compute the dynamical potential $V_{\mathrm{dyn,k}}(\phi)$, and thus the Schwinger functional $W[J]$, in the complex plane. Convergence is achieved until some critical value of the current $J_c$. For $|J_y| > J_c$ the numerical solution displays a *blow-up*, i.e. it develops a singularity at some finite $k > 0$. This possibility was discussed in Section 4.1. The existence

Table 4: Fit parameters to the fit in (F.2) and plotted with the data-set in Figure 11.
The error to the fit parameters in $d = 4$ is given by removing/adding the last 10 data-
points from/to the fit. The $d = 0$ fit only uses the data-points left of the singularity.
Here, errors were obtained by adding 25 data-points to the fit interval (generated
from the numerical integral). Surprisingly, we find that the fit allows for big devia-
tions in the parameters $a, b$ and $c$. The position of the singularity, however, is barely
affected by adding/removing data points from the fit.

| Fit param. | $a$ | $b$ | $J_c$ | $c$ |
|---|---|---|---|---|
| $d = 0$ | 0.388(90) | 1.0008(55) | 3.002104(65) | −0.281(12) |
| $d = 4$ | 9.7(88) | 1.24(41) | 3.3659(42) | −0.99(91) |

of complex poles in the potential of the Schwinger functional follows from the analytic struc-
ture of the equation, see [64, 65]. In $d = 0$ the blow up can directly be associated with the
zero of the generating functional. In any case, the numerical blow-up is the final slice, for
which the expansion around $\phi_0$ in the RG-adapted scheme still applies. The last converging
computation in $d = 4$ is at $\phi_y = 3.25$ ($J \approx 3.35$). Although we are not at the pole position yet,
there is a premature blow up in the equations due to the numerical approximation scheme.
The occurrence of oscillations around discontinuities is expected in numerical schemes using
polynomials and is generally accounted for in the numerical fluxes. In the close proximity of
a blow up, the arising oscillations create negative diffusion, which causes an instant failure
of the computation, or prematurely triggers a blow-up. These issues can be dealt with using
positivity preserving LDG-schemes, see [50], but is not within the scope of this work.

The converged slices are interpolated in $\phi_y$ direction. To resolve the real direction we use
a grid from $\phi_x \in [−3, 3]$ with a polynomial order $N = 2$ and a cell number $K = 200$, to further
reduce the occurrence of oscillatory behaviour. In Figure 11 we show the raw data from the
evaluation of the RG-adapted flow (56). The build up of the singularity is clearly visible, all
other structures of the potential vanish in comparison.

We are now interested in obtaining the exact position of the divergence from the derivative
of the effective interaction potential. We make use of the Cauchy-Riemann equations to obtain
the expression for the real part of the dynamical potential,

$$\text{Re}[V_{\text{dyn}}(0, \phi_y)] = \int_0^{\phi_y} d\phi_y' \, \frac{\partial \text{Re}[V_{\text{dyn}}]}{\partial \phi_y'}(0, \phi_y')$$

$$= -\int_0^{\phi_y} d\phi_y' \, \frac{\partial \text{Im}[V_{\text{dyn}}]}{\partial \phi_x'}(0, \phi_y'). \tag{F.1}$$

Furthermore, we obtain the full potential of the Schwinger functional from (61). Since we
are investigating a zero of the generating functional, we use a logarithm as a fit function.
To accommodate the overall structure of the potential, we add a mass parameter $c$, which
significantly improves the stability of the fit. This choice is further supported by previous
analyses [64], which suggest a simple, purely imaginary pole in the first derivative of the
potential. The fit-function is given by,

$$\text{Re}[w(0, J_y)] = c \, J_y^2 + a + b \log(J_c − J_y), \tag{F.2}$$

where $J_c$ is the position of the singularity. To check if this function is able to fit the position
correctly, we perform a benchmark check in $d = 0$, additionally to the $d = 4$ data. In $d = 0$ we

evaluate the integral (4) directly, as discussed in Section 5.3. The fit parameters are given in Table 4. The fit error is determined by adding/removing data points from the fit-interval, as indicated in Figure 11. The blow-up is located at $J_c = 3.3659(42)i$ and contains only a very small error from the fit.

# G   Polchinski flow

Additionally to the RG-adapted scheme, we explore the approach using a classical propagator in Appendix A, which is simply the Polchinski flow. In this scheme, the effective interaction potential also carries the change to the classical mass term $V_{\text{int,k}}(\phi) = V_{\text{dyn,k}}(\phi) + \frac{m^2 - m_k^2}{2!}\phi^2$. Following the derivation we define the current via the classical propagator at the expansion point $\phi_0 = 0$, to wit

$$J = \left(G_k^{(0)}[\phi_0]\right)^{-1}\phi = \left(m^2 + p^2 + R_k(p^2)\right)\phi. \tag{G.1}$$

The insertion in (A.84) yields:

$$\partial_t V_{\text{int,k}}(\phi) = \frac{1}{2}\int_p \left\{ \frac{\left(\left(V_{\text{int,k}}^{(1)}(\phi)\right)^2 - V_{\text{int,k}}^{(2)}(\phi)\right)\partial_t R_k(p^2)}{(m^2 + p^2 + R_k(p^2))^2} \right\}$$

$$= v(d)\frac{k^{2+d}}{(m^2 + k^2)^2}\left[\left(V_{\text{int,k}}^{(1)}(\phi)\right)^2 - V_{\text{int,k}}^{(2)}(\phi)\right], \tag{G.2}$$

where the trace is evaluated over momentum space. Additionally, a flat cutoff is used, for details see Appendix L.

The choice $\phi_0 = 0$ ensures the prefactor $\frac{k^2}{(m^2+k^2)^2}$ is real and positive, which is not trivial for arbitrary complex fields $\phi_0$. Since it is the simplest numerical scenario, we present results for the $\phi_0 = 0$ case. For the sake of comparison we use the same initial conditions as given in Section 5.1.2. With the classical mass $m^2 = 1$, we have $J_k = (1+k^2)\phi$, i.e. $J_0 = \phi$ at $k = 0$.

Computations are performed on slices of constant $\phi_y$. To this aim we use:

- Values of $\phi_y$ spaced by 0.1.

- A 1d-numerical grid, ranging from $\phi_x \in [-3,3]$ containing $K = 60$ cells and a polynomial of order $N = 2$ in each cell.

Figure 1 shows the comparison to the analytic result. The current cell density allows to resolve the potential up to $\phi_y = 2.4$, the pole is situated at $\phi_y = 3$. Numerical computations break down due to the pole building up in the numerical expressions for (50). This can be improved slightly by increasing the cell density of the grid, but not infinitely.

Naively, we could try to improve the numerical scheme by chose a different expansion point, closer to the pole position. However, in Section 4 we discussed the necessity of real, positive diffusion $a(t)$ in (44) to ensure numerical stability and convergence. For arbitrary $\phi_0$ this is no longer ensured. By reading off of (1) we obtain,

$$0 \overset{!}{\leq} a(k) = \frac{k^2}{\left[S_k^{(2)}\right]^2} = \frac{k^2}{\left(m^2 + k^2 + \frac{\lambda}{2}\phi_0^2\right)^2}. \tag{G.3}$$

It follows that $\left(S_k^{(2)}\right)^2 \in \mathbb{R}_{>0}$ restricts the choice of $\phi_0$ to the coordinate axes. This is additionally enforced by the assumption made in Section 4.3, which explicitly prohibits introducing a

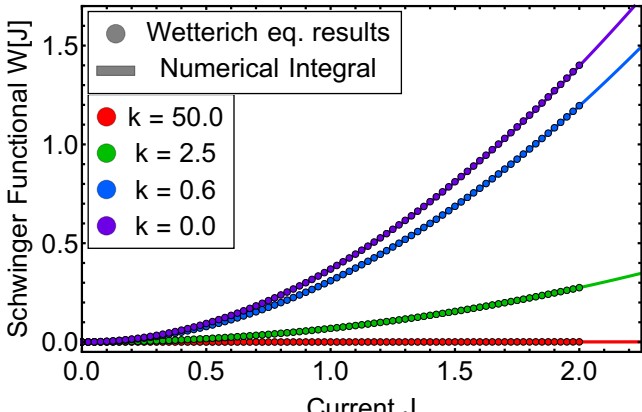

Figure 12: RG-time dependence of the Schwinger functional computed from the 1PI flow and by a direct numerical evaluation in $d = 0$. All units are given in terms of the UV mass $m = 1$ with an initial cutoff $\Lambda = 5$.

dependency on an additional complex variable. Choosing an expansion point along the imaginary axis $\phi_0 = i\,|\phi_0|$ with $|\phi_0^2| > 2\frac{m^2}{\lambda}$ eventually introduces a pole to the flow and is therefore also impractical for numerical computations.

## H 1PI flow

In this section we discuss the 1PI flow in a complex setting. This is illustrated by the example of the numerical test case $d = 0$, which is also used as a benchmark for numerical convergence of the Wilsonian effective action flows in Section 5.2. We make use of LPA 1PI effective action in (42). The full propagator is then given by

$$G_k[\phi] = p^2 + V_{\mathrm{eff},k}^{(2)}(\phi) + R_k(p^2). \tag{H.1}$$

This expression is inserted into the 1PI flow (29) using a flat Regulator, see Appendix L. Evaluating the momentum trace yields the known flow of the effective potential $V_{\mathrm{eff},k}$ in an $O(1)$ theory [16, 66],

$$\partial_t V_{\mathrm{eff},k}(\phi) = v(d)\frac{k^{d+2}}{k^2 + \partial_\phi^2 V_{\mathrm{eff},k}(\phi)}. \tag{H.2}$$

Equation (H.2) is rewritten in terms of the second derivative of the potential $u(\phi) = \partial_\phi^2 V_{\mathrm{eff},k}(\phi)$, to suit the numerical framework presented in Section 4.2 and more thoroughly in [46]

$$\partial_t u = -\partial_\phi\Big(\frac{v(d)k^{d+2}}{(k^2 + u)^2}\partial_\phi u\Big) = -\partial_\phi\Big(\frac{v(d)k^{d+2}}{k^2 + u}p\Big),$$

$$p = \partial_\phi\Big(\log(1 + u/k^2)\Big), \tag{H.3}$$

where $p_k(\phi)$ is solved in a separate, stationary equation. For an in depth discussion of the numerical framework in the context of 1PI flows see [46], in this work we only focus on the extension to the complex plane. The positivity condition for the real diffusion term follows from the square in the denominator. Since the equation is exact in $d = 0$ we can recover the RG-running of the Schwinger functional by performing the (modified) Legendre transformation (10), see Figure 12 in the case $d = 0$.

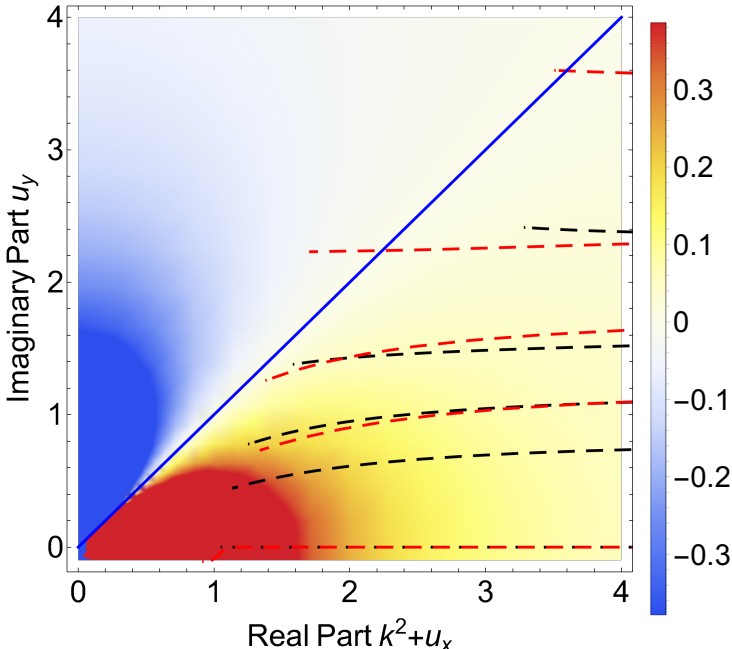

Figure 13: Real part of the diffusion $\mathrm{Re}[f] = f_x$ in a complex setting. The blue line indicates $\mathrm{Re}[f] = 0$. The dotted lines follow the diffusion flux for different $q_x = 0, 0.8, 1.2, 1.6, 1.8$ at $q_y = 1.0$ (black) and $q_y = 1.5$ (red). The trajectories start at the right side of the plot and move to the left as the RG-scale $k$ decreases. Whilst the diffusion is positive to the right of the blue line, trajectories crossing this line become unstable. The black lines are therefore stable. The computation of the red lines is numerically unstable and breaks off at $k \approx 0.85$, when the blue line is crossed for a specific $q_x$ and the diffusion is no longer positive. All units are given in terms of the UV mass $m = 1$ with an initial cutoff $\Lambda = 5$.

## H.1 The complex 1PI flow

Following the arguments in Section 4.3, the 1PI flow is now extended to the complex plane. For complex values, (H.3) is a system of four equations. The substitution $\partial_{\bar{z}} \to \partial_x$, where $z = x + \mathrm{i}\, y$ is performed, to wit

$$\partial_t u_x + \partial_x \left( f_x \partial_x u_x - f_y \partial_x u_y \right) = 0, \tag{H.4}$$

$$\partial_t u_y + \partial_x \left( f_x \partial_x u_y + f_y \partial_x u_x \right) = 0, \tag{H.5}$$

and the flux $f = v(d)k^{d+2}/(k^2 + u)^2$, compare (H.3). The direct comparison to (49) and (50) shows the increased complexity of the equations. Most importantly, the flux $f$ is no longer necessarily strictly positive in a complex setting.

Before attempting to naively solve this system with the given framework, its stability within the presented numerical scheme needs to be investigated. General requirements for stability of complex partial differential equations are known and have been studied in the context of Schrœdinger-type equations, or the image-denoising process [67, 68]. In fact, stability of numerical schemes can be proven for certain classes of non-linear complex diffusion equations. The most central elements of this proof are the positivity of $f_x$ and finiteness of the flux. The second requirement is quite straight-forward, and we briefly explain the former by the example of a Gaußian: While convection movements will move the mean of the Gaußian, positive diffusion is known for smoothing out the structure, i.e. increasing the width of the Gaußian as (RG-)time progresses, see Figure 16a. Negative diffusion can be thought of as

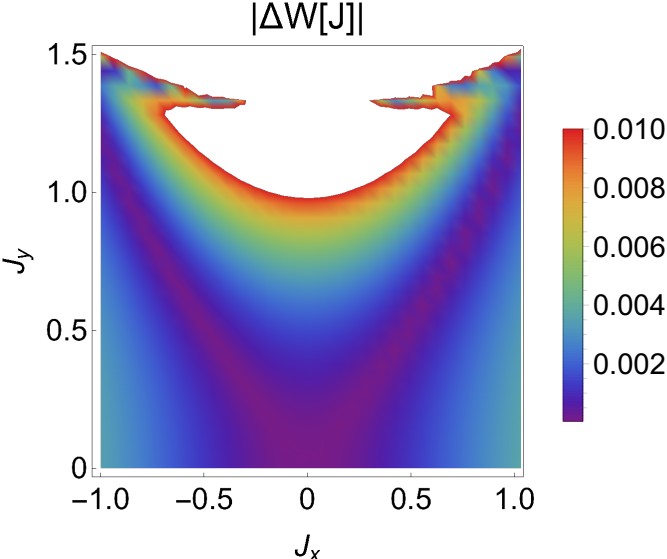

Figure 14: Absolute, relative error on the Legendre transformed of the 1 PI effective Potential $V_{\text{eff}}$ in $d = 0$. The numerical evaluation of the integral in (4) is used as a reference. All units are given in terms of the UV mass $m = 1$ with an initial cutoff $\Lambda = 5$.

simply switching the sign in the time evolution: Instead of increasing the width, it is decreased. As the (RG-)time evolution progresses, an uncontrollably growing Gaußian develops, which is highly numerically instable. Nevertheless, systems containing two-way diffusion, i.e. diffusion of varying sign, have been studied in literature [69,70]. Numerical solutions to purely diffusive systems containing two-way diffusion rely on iterative procedures. Such algorithms require previous knowledge of the solution at the end $t_1$ and beginning $t_0$ of the time-integration [71].

In case of the 1PI flow, the finiteness of the diffusive flux is ensured by the convexity restoring properties of the 1PI flow. The pole in the flow (H.3) is never reached [72]. However, $f_x + \mathrm{i} f_y = \frac{k}{(k^2 + u_x + \mathrm{i} u_y)^2}$ does not ensure positive diffusion as it does on the real axis, in fact, it is spoiled by it. To illustrate this we perform a computation at different field values and track the real diffusion throughout our calculation Figure 13. This plot shows the numerical limitations when trying to solve the complex 1PI flow. We find that with increasing external imaginary field, also the trajectories with negative diffusion increase and calculations become unstable. We have yet to determine if this is caused by a numerical fine-tuning problem or is a property of the flow itself. Diffusive contributions remain positive until $\phi_y = 1$, which coincides with the initial UV-mass.

Finally, the full expression for the 1PI effective potential is obtained by the integration procedure outlined in Appendix J.

## H.2 The complex Legendre transform

In this Section we discuss the complex Legendre transformation of a holomorphic function. The difficulty in defining a general complex Legendre transformation is within the definition of the derivative. For a holomorphic function this issue is resolved, since derivatives are well defined. In the present case, we make use of the mapping $\partial_z \to \partial_x$ for a derivative with respect to the complex number $z = x + \mathrm{i} y$, which was discussed in Section 4.3. Then, the coordinate

transformation can be inferred from the first derivative, using

$$J = \frac{\partial V_{\text{eff}}}{\partial \phi_x}[\phi_x, \phi_y] \quad \Rightarrow \quad \phi(J_x, J_y) = \left(\frac{\partial V_{\text{eff}}}{\partial \phi_x}\right)^{-1}[J_x, J_y].\tag{H.6}$$

Now, the Legendre transformation is naturally extended to the complex plane by

$$W[J_x, J_y] = \left(J_x + \mathrm{i}J_y\right)\phi(J_x, J_y) - \Gamma\left[\phi(J_x, J_y)\right].\tag{H.7}$$

The error on the Legendre transformation is computed by a comparison with the direct numerical evaluation of the integral in (4). We compute the absolute relative error from,

$$|\Delta W[J]| = \left|\frac{W_{\text{1PI}} - W_{\text{int}}}{W_{\text{1PI}}}\right|[J].\tag{H.8}$$

The result is shown in Figure 14. Furthermore, we obtain $V_{\text{dyn}}$ from the 1PI Schwinger functional $W_{\text{1PI}}$ using (41). It convergence, in comparison to the RG-adapted scheme and the Polchinski flow is shown in Figure 1. We conclude, that the current numerical evaluation of the 1PI flow needs further improvement for evaluations in the complex plane.

# I  Analytic relations for the susceptibility

The susceptibility indicates the response of the magnetisation $M$ to the application of an external field $H$ and is given by

$$\chi(\phi_y, H = 0) = \frac{\partial M}{\partial H}(\phi_y, H)|_{H=0}.\tag{I.1}$$

The magnetisation is expected to diverge at the phase transition following (68). Hence, a direct, numerical evaluation of this expression is impractical. Therefore, we make use of the following analytic relation. We start from the definition of the magnetisation, via the equation of motion (63b), by taking a derivative with respect to the external field $H$.

$$1 = \partial_H \frac{\partial w[J(\phi)]}{\partial \phi}\bigg|_{\phi=\phi_{\text{EoM}}}$$
$$= \frac{\partial \phi_{\text{EoM}}}{\partial H} \frac{\partial^2 w[J(\phi)]}{\partial \phi^2}\bigg|_{\phi=\phi_{\text{EoM}}},\tag{I.2}$$

where $\phi_{\text{EoM}} = M$ is simply the magnetisation, see (63b). Furthermore, $\partial_\phi^2 V_{\text{dyn}}$ is directly evaluated in the LDG-scheme (see Section 4.4) and relates to the effective potential of the Schwinger functional via (61). Putting everything together, the susceptibility can be evaluated via

$$\chi(\phi_y, H = 0) = \frac{1}{\left(m_{k=0}^2 - \partial_\phi^2 V_{\text{dyn}}(\phi)\right)}\bigg|_{\phi=\phi_{\text{EoM}}},\tag{I.3}$$

where $m_{k=0}^2$ is the RG-time dependent mass at $k = 0$, depicted in Figure 5. Equation (I.3) forgoes taking a difficult numerical derivative with respect to $H$. An additional benefit is easy access to the divergence, via the zeroes of the enumerator.

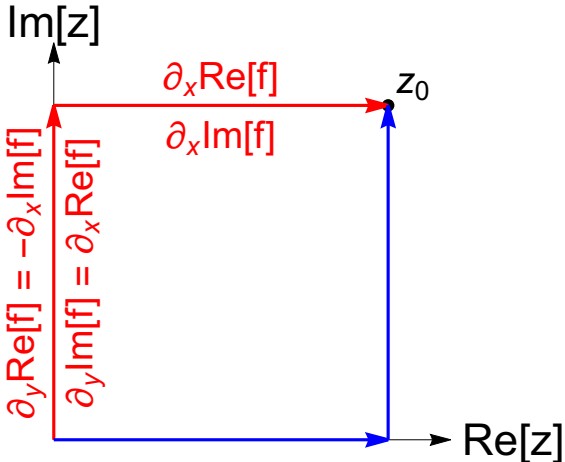

Figure 15: Path choices for an integration of the holomorphic function $f$ at some co-ordinate $z_0$ in the complex plane. Both the red and blue lines are convenient choices, since the computation gives $\partial_x \text{Re}[f]$ and $\partial_x \text{Im}[f]$ as an output.

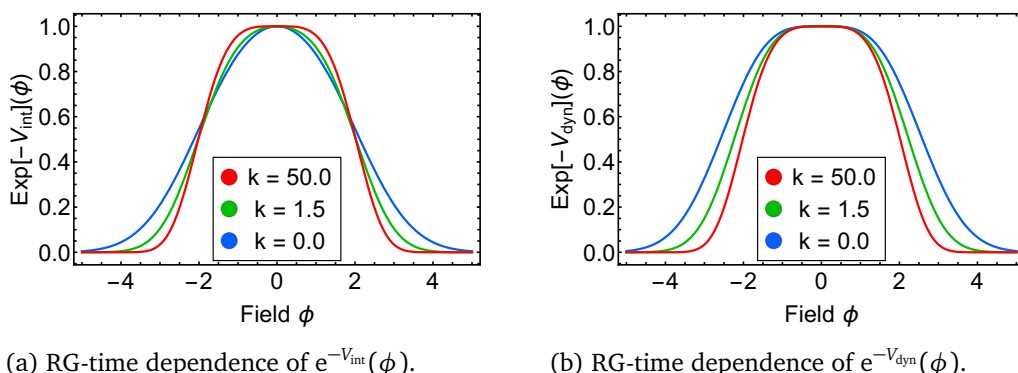

(a) RG-time dependence of $e^{-V_{\text{int}}}(\phi)$.

(b) RG-time dependence of $e^{-V_{\text{dyn}}}(\phi)$.

Figure 16: Comparison of the exponential formulations of the Polchinski flow (K.1) and the RG-adapted scheme (K.3) in $d = 0$. One can clearly see the development of a $\phi^2$ contribution to the exponential in (a) and the lack thereof in (b), due to the RG-adapted scheme. All units are given in terms of the UV mass $m = 1$ with an initial cutoff $\Lambda = 5$.

## J  Complex integration and holomorphicity

In this Section we discuss how we obtain the full potential from the numerical data. For an explicit complex integration a convenient path in the complex plane can, and should, be chosen due to holomorphicity, which we implemented in Section 4.3. Possible path choices are indicated in Figure 15.

First, we check if our results are holomorphic: to this aim we integrate along the red and blue path in Figure 15 and compute their relative error given by

$$\Delta I = \frac{I_{\text{red}} - I_{\text{blue}}}{I_{\text{red}}}. \tag{J.1}$$

The Cauchy-Riemann equations are used to obtain the respective $y$ derivatives. The relative error remains below $10^{-3}$ for all schemes up until $\phi = 2$. From there on, a small numerical error is generated, which stems from the interpolation of steep structures in post-processing.

Generally, all integrated results are obtained by integration along the red path. Although ideally both paths should yield the same result, the numerical precision along the red path is

Table 5: Absolute value of the difference in solutions $\Delta e^{-V_x}$, computed by the exponential formulation and the formulation in $V_{\text{dyn/int}}$, see Section 5 and Appendix G. The error is computed using (K.5). We take this as an estimate of the error generated by the flux-boundary conditions.

| Dimension | 0 | 1 | 2 |
|---|---|---|---|
| Classical Prop. | $2.5 \ 10^{-4}$ | $3.1 \ 10^{-4}$ | $4.4 \ 10^{-4}$ |
| RG-adapted Prop. | $5.9 \ 10^{-5}$ | $5.9 \ 10^{-5}$ | $6.0 \ 10^{-5}$ |

superior: The numerical grid, and thus the high numerical precision, follows the real direction. Therefore, the biggest contribution to the path should follow from a horizontal line, whereas the vertical contribution should, preferably, be small. The red path is now favoured by two observations:

- Symmetry dictates, that either the real or imaginary part of the function $f$ is zero on the imaginary axis, i.e. the vertical red path.

- The imaginary part on the real axis is zero, i.e the horizontal, imaginary blue path does not contribute at all.

## K  Alternative formulations

The Polchinski equation is very challenging to resolve numerically. One specific challenge is the determination of boundary conditions, since the flow of the Wilsonian effective action increases at higher field values. This is due to many remaining redundancies which are removed in the 1PI effective action.

To circumvent this problem and to quantify the numerical error at the boundaries, we make use of an alternative formulation in terms of exponentials. Structurally, these new equations resemble Wegner's flow (12), whose numerical properties are discussed in Section 4.1.1. This has the additional advantage that we can directly solve for the exponentials of $S_{\text{int},k}[\phi]$, (A.83), and $S_{\text{dyn},k}[\phi]$, (35), instead of using the first derivative. However, using this formulation is increasingly challenging at higher imaginary field values. A bigger imaginary part of $\phi$ ia accompanied by increasingly strong oscillations. Resolving the initial conditions already requires a very high cell density. So whilst this formulation is not tailored to perform computations at big imaginary fields, we find that it makes for a useful check of convergence close to the real axis.

First we focus on reformulating the Polchinski flow (A.84) in terms of $\exp(-S_{\text{int},k}[\phi])$. Multiplying the equation with $-\exp(-S_{\text{int},k}[\phi])$ yields

$$\partial_t e^{-S_{\text{int},k}[\phi]} = \frac{1}{2} \text{Tr} \ \partial_t G_k^{(0)} \left[ \frac{\delta^2}{\delta\phi\delta\phi} e^{-S_{\text{int},k}[\phi]} \right] - \frac{1}{2} G_k^{(0)} \partial_t S_k^{(2)} e^{-S_{\text{int},k}[\phi]}. \qquad \text{(K.1)}$$

To simplify the equation further we replace $e^{-S_{\text{int},k}[\phi]} \to N_k e^{-S_{\text{int},k}[\phi]}$, where $N_k$ is an RG-time dependent constant in $\phi$. Introducing $N_k$ does not affect physics, since the physical correlation functions are normalised by definition. This results in

$$\partial_t e^{-S_{\text{int},k}[\phi]} = A(k) \left[ \frac{\delta^2}{\delta\phi\delta\phi} e^{-S_{\text{int},k}[\phi]} \right] - B(k) \ e^{-S_{\text{int},k}[\phi]} - \partial_t(\ln(N_k)) e^{-S_{\text{int},k}[\phi]}, \qquad \text{(K.2)}$$

after dividing by $N_k$ and $A, B$ can be read off of (K.1). Now we discuss possible options to determine $N_k$: the obvious choice would be fixing some normalisation, i.e. $\int d\phi \, N_k e^{-S_{\text{int},k}}[\phi] = 1$. Another, numerically much more convenient choice would be to impose $\partial_t(\ln(N_k)) = -B(k)$, and thereby dropping the last line of (A.84). We chose the normalisation $N_k$ such that $S_{\text{int}}[0] = 0$.

Let us now consider the RG-adapted flow (39a). Multiplying by $e^{-S_{\text{dyn},k}}[\phi]$ yields

$$\left(\partial_t + \int_x \phi \, \gamma_{\text{dyn},k} \frac{\delta}{\delta \phi}\right) e^{-S_{\text{dyn},k}}[\phi] - \frac{1}{2} \int_x \phi \, \partial_t \Gamma_k^{(2)}[\phi_0] \phi \, e^{-S_{\text{dyn},k}}[\phi]$$

$$= \frac{1}{2} \operatorname{Tr} \mathcal{C}_k \left[\frac{\delta^2}{\delta \phi \delta \phi} e^{-S_{\text{dyn},k}}[\phi]\right]. \qquad (K.3)$$

Again, all terms $\propto e^{-S_{\text{dyn},k}}[\phi]$ can be dropped by introducing an appropriate normalising condition. The RG kernel $\mathcal{C}$ is given by the RG-adapted kernel (30b).

The flows are now solved in terms of the potentials $V_{\text{int}}$ and $V_{\text{dyn}}$. Their relation to the interaction part $S_{\text{int}}$ and the dynamical part $S_{\text{dyn}}$ of the effective action follows immediately from their respective definitions (41) and (40), to wit

$$V_{(\text{int}/\text{dyn})} = S_{(\text{int}/\text{dyn})}/\mathcal{V}_d \quad \text{and} \quad \mathcal{V}_d = \int d^d x. \qquad (K.4)$$

In our approximation, the volume $\mathcal{V}_d$ drops out of the equations. Finally, (K.1) and (K.3) are solved for varying dimensions $d = 0$, to $d = 2$ on the real axis. The results in $d = 0$ are depicted in Figure 16. Figure 16a clearly shows the development of a $\phi^2$ contribution to $V_{\text{int}}$ in the Polchinski flow. In contrast, Figure 16b demonstrates the lack of a $\phi^2$ contribution to $V_{\text{dyn}}$ in the RG-adapted scheme. This is in accordance with (37) and beautifully emphasises the reduced degree of redundancy of the scheme. Table 5 gives the error $\Delta e^{-V_x}$ between solutions computed by the exponential formulation and the formulation in $V_{(\text{int}/\text{dyn})}$ at $\phi = 2$, to wit

$$\Delta e^{-V_x} = e_h^{-V_x}(\phi) - e^{-V_{x,h}(\phi)}|_{\phi=2}, \qquad (K.5)$$

where $x = \text{int}/\text{dyn}$. The index $h$ indicates the numerically computed object. The field value $\phi = 2$ is chosen, because it displays the most dynamics in the RG-time evolution and also generates the biggest overall error. The error values in Table 5 are satisfyingly small, with a slight superiority of the RG-adapted scheme.

## L  Regulator

In the present work we use the 4-dimensional flat or Litim regulator, see [33]. For a scalar, the regulator is given by

$$R_k(p) = p^2 r(x), \quad r(x) = \left(\frac{1}{x} - 1\right) \theta(1 - x), \qquad (L.1)$$

with $x = p^2/k^2$. The flat cutoff is optimised for the 0th order in the derivative expansion, which is used throughout this work, [13, 31, 32].

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
