# Peer review of "Functional flows for complex effective actions"

_SciPost Physics, doi:SciPost Phys. 15, 074 (2023)_

## Round 1 · Referee Report · Anonymous (Referee 1) · 2023-4-8

Report

The authors have satisfactorily responded to the comments and questions of my first report. I recommend publication of the manuscript in the present form.

---

## Round 1 · Author Response

We thank the referees for their remarks and suggestions. The referees addressed two main questions:

1) What about out of equilibrium, complex action problems/open quantum systems?

The RG flows derived in the present manuscript are applicable to more general complex problems. This is a technical upgrade, a reduction of the number of coupled differential equations specific to theories with only a complex source does not hold anymore. We have added respective comments in the introduction (second paragraph, starting with 'The explicit applications ....') and picked up this point again in Section IV C on page 10, first paragraph, in which we use the reduction. The performance of the approach in a more general situation certainly depends on the details of the complex structure, but these are more technical, and not conceptual obstructions.
We have added references to more general complex problems.

2) Why does the 1PI flow perform so badly?

Numerically, the failure of the 1PI flow is linked to the non-algebraic structure of the equation, which is explained in App. H, and is linked to the real part of (G)^2 ((propagator squared) becoming negative at finite RG-scale k. We have added a reference to App. H to the description of Fig 1 (a) for clarification. In our current understanding this problem can be overcome within a full RG-adaptation of the flow, including an adapted choice of the initial condition (the latter choice can be used to avoid flows that cross these singularities). The intricacy described above is directly linked to the non-algebraic structure of 1PI flows, it is absent in flows for the Wilsonian effective action, hence the differences within the rather hands-on application of all these flows in the present work.

In short, we believe that the relative shortcomings of the 1PI flows are overcome in updates of the approach along the lines indicated above. While we have already done quite some work in this direction, we believe that our understanding has not matured enough to add more specific comments in the draft. However, we hope to report on this matter soon.

---

## Round 1 · List of Changes

-SUMMARY OF CHANGES-

-Introduction, second paragraph, clarification on point 1):

The explicit applications in this work concentrate on the case complex sources, related to the Lee-Yang zeros mentioned above. We emphasise that the present complex action approach also accommodates quantum field theories with generic complex couplings. While also highly interesting e.g. in the context of PT-symmetric theories, explicit applications in this area will be discussed elsewhere. Here we only briefly discuss the minor technical differences within the numerical implementation of such flows.

-Section II, third paragraph, references to more general complex problems (concerning point 1)):

The more general situation, which occurs for example in applications to open quantum systems or PT-symmetric theories \cite{doi:10.1063/1.5115323, Grunwald:2022kts}, will be discussed elsewhere.

-Section IV C, first paragraph, clarification on point 1):

The present set-up and in particular the final flow equation \labelcref{eq:FlowPol} can be readily applied to complex action problems with general complex couplings. Then, \labelcref{eq:FlowPol} is a partial differential equation for a general complex function $u(z)$ of a complex variable $z$. For complex classical couplings we find
%
\begin{align}
\overline{u(z)} \neq u(z)\,.
\label{eq:GenericCase}
\end{align}
%
Accordingly, \labelcref{eq:FlowPol} has to be solved for its imaginary and real part, while also implementing the holomorphicity constraint $\partial_{\bar z} u(z) =0$.

From now on we restrict ourselves to the case of complex currents. Then, \labelcref{eq:FlowPol} is a partial differential equation in the RG-time t and the complex spatial variable $z$. In principle, one could simply use the split of $z$ into its real and imaginary part, $z= x + \imag \ y $, and the respective split of the derivative, $ \partial_z = \frac{1}{2}(\partial_x - \imag \ \partial_y) $, for solving the equation on a two-dimensional grid. However, this is a two-dimensional representation of a one-dimensional system, additionally necessitating the implementation of the holomorphicity constraint.

-Section V, Figure 1, we have added a reference to App H (concerning point 2)).

-Section VII (Summary), second paragraph, we have added a comment on point 2):

The numerical results for Lee-Yang zeroes have been obtained from RG-adapted flows of the Wilsonian effective action. In turn, the naive implementation of the commonly used 1PI flow has a much smaller convergence radius in the complex plane. In our opinion, a fully RG-adapted 1PI flow based on \labelcref{eq:GenFlow1PI} as well as a careful choice of the initial condition should resolve this issue, and we hope to report on this matter in the near future.

---

## Editorial Decision

published